# Qualitative Evaluation of a Quality Improvement Collaborative Implementation to Improve Acute Ischemic Stroke Treatment in Nova Scotia, Canada

**DOI:** 10.3390/healthcare12181801

**Published:** 2024-09-10

**Authors:** Shadi Aljendi, Kelly J. Mrklas, Noreen Kamal

**Affiliations:** 1Faculty of Computer Science, University of New Brunswick, Fredericton, NB E3B 5A3, Canada; 2Department of Industrial Engineering, Dalhousie University, Halifax, NS B3J 1B6, Canada; noreen.kamal@dal.ca; 3Strategic Clinical Networks™, Provincial Clinical Excellence, Alberta Health Services, Edmonton, AB T5J 3E4, Canada; kelly.mrklas@albertahealthservices.ca; 4Department of Community Health and Epidemiology, Faculty of Medicine, Dalhousie University, Halifax, NS B3H 1V7, Canada; 5Department of Medicine (Neurology), Dalhousie University, Halifax, NS B3H 3A7, Canada

**Keywords:** acute ischemic stroke, quality improvement collaborative, qualitative analysis, barriers and facilitators, healthcare implementation, CFIR

## Abstract

The Atlantic Canada Together Enhancing Acute Stroke Treatment (ACTEAST) project is a modified quality improvement collaborative (mQIC) designed to improve ischemic stroke treatment rates and efficiency in Atlantic Canada. This study evaluated the implementation of the mQIC in Nova Scotia using qualitative methods. The mQIC spanned 6 months, including two learning sessions, webinars, and a per-site virtual visit. The learning sessions featured presentations about the project and the improvement efforts at some sites. Each session included an action planning period where the participants planned for the implementation efforts over the following 2 to 4 months, called “action periods”. Eleven hospitals and Emergency Health Services (EHS) of Nova Scotia participated. The Consolidated Framework for Implementation Research (CFIR) was utilized to develop a semi-structured interview guide to uncover barriers and facilitators to mQIC’s implementation. Interviews were conducted with 14 healthcare professionals from 10 entities, generating 458 references coded into 28 CFIR constructs. The interviews started on 17 June 2021, 2 months after the intervention period, and ended on 7 October 2021. Notably, 84% of these references were positively framed as facilitators., highlighting the various aspects of the mQIC and its context that supported successful implementation. These facilitators encompassed factors such as networks and communications, strong leadership engagement, and a collaborative culture. Significant barriers included resource availability, relative priorities, communication challenges, and engaging key stakeholders. Some barriers were prominent during specific phases. The study provides insights into quality improvement initiatives in stroke care, reflecting the generally positive opinions of the interviewees regarding the mQIC. While the quantitative analysis is still ongoing, this study highlights the importance of addressing context-specific barriers and leveraging the identified facilitators for successful implementation.

## 1. Introduction

Significant advances leading to reductions in patient disability have been made in ischemic stroke treatment over the past two to three decades. Alteplase, a tissue plasminogen activator, has been demonstrated to be effective in dissolving blood clots and restoring blood flow when administered early, thereby reducing long-term disability in stroke patients since its introduction in the mid-1990s [1]. Additionally, mechanical treatment with endovascular thrombectomy (EVT) has been found to be beneficial in improving functional outcomes and reducing mortality in stroke cases, as evidenced by a series of randomized clinical trials in 2015 [2]. However, the effectiveness of both treatments is highly time-dependent [3,4], as treatment with alteplase should begin within 30 min (median) of arrival at the hospital [5,6], and EVT should be started within 60 min (median) from hospital arrival. Treatment of acute ischemic stroke (AIS) with both treatments is a complex process integrating the efforts of various hospital departments and interdisciplinary healthcare professionals. Improving stroke treatment’s efficiency [7,8] and increasing the proportion of ischemic stroke patients that receive treatment [9,10,11] have been effortful endeavours for several decades.

Quality improvement collaboratives (QICs) [12] are widely used to improve healthcare processes and patient outcomes. The first steps in quality improvement collaboratives were taken by the Northern New England Cardiovascular Disease Study Group in 1986 and the Vermont Oxford Network in 1988 [13]. The Institute for Healthcare Improvement added to this in 1995 with its breakthrough series [14]. The purpose of QICs is to make improvements across multiple facilities or, specifically in this study, across multiple hospitals. The original Institute for Healthcare Improvement (IHI) white paper on the Breakthrough Series Collaborative Model had specified a 1-year duration; however, in practice, the duration of this model varies. The year-long project consists of the following: recruitment of improvement teams at each site, three 2-day learning sessions, three action periods, and a closing congress. The action periods follow each learning session, which are spaced 4 months apart. Several publications have addressed the effectiveness of QICs [12,15]. The QIC methodology has previously been used to improve acute stroke treatment processes with varying success [16,17,18].

The Atlantic Canada Together Enhancing Acute Stroke Treatment (ACTEAST) project [19] is a modified quality improvement collaborative (mQIC) aimed at improving ischemic stroke treatment rates and time-based efficiency across Atlantic Canada. The project is considered a modified QIC due to several adjustments. Notably, the improvement collaborative’s duration was shortened from the typical 1 year to 6 months. Consequently, there were only two learning sessions and action periods, whereas the usual format includes three. Furthermore, all the project’s aspects were conducted entirely virtually. This project included all Atlantic provinces but began in Nova Scotia, where the initiative was first launched. Nova Scotia was targeted for this initial evaluation because it is the largest province in the region and has the highest number of stroke treatment centers. This study evaluated the efficacy of the mQIC in Nova Scotia (Canada) using the Consolidated Framework for Implementation Research (CFIR) [20], which provided a clear framework for the evaluation. The primary research question guiding this study was “What are the barriers and facilitators within each CFIR construct reported by the participants of an mQIC during an initiative to improve the access and time efficiency of treatment of acute ischemic stroke patients with thrombolysis and EVT in the Canadian province of Nova Scotia?” Correspondingly, the aim of this study was to identify and analyze the barriers and facilitators within the CFIR construct as reported by the participants involved in an mQIC to enhance access and time efficiency in the treatment of acute ischemic stroke patients with thrombolysis and EVT in Nova Scotia, Canada.

## 2. Materials and Methods

### 2.1. Study Design

Nova Scotia has a well-established stroke system of care. The province has designated stroke centers distributed across the province to ensure adequate access. Furthermore, they have support by way of a designated stroke coordinator for each hospital and a provincial stroke registry to collect key data on stroke care in the province.

The ACTEAST project primarily aimed to enhance the efficiency of AIS treatment in the Canadian Atlantic provinces by increasing the proportion of patients receiving Tissue Plasminogen Activator (tPA) and/or EVT by 5% and reducing door-to-needle time (DTN) by 15 min. 

This project used a mQIC that enrolled all stroke hospitals in the Canadian province of Nova Scotia. The mQIC for Nova Scotia has been described in detail previously [21]. Briefly, the mQIC was 6 months long, and prior to the start of the mQIC, all stroke hospitals in the province assembled and enrolled teams to the mQIC; the teams were made of representatives from key professions involved in the treatment process, including emergency physicians, emergency department (ED) nurses, radiologists, computed tomography (CT) technologists, administrators, paramedics, and stroke coordinators. There were two full-day learning sessions, and the mQIC began at the first learning session and the second learning session was held approximately 2 months after the first one. The first learning session focused on presenting the evidence for alteplase treatment, EVT, and imaging for acute stroke, and the second learning session focused on hearing about improvements from the participating teams. Both learning sessions provided time for each team to plan their changes (action planning), which was reported back to the entire group. The teams were supported throughout the 6 months with site visits and webinars.

Nova Scotia is a small Canadian province of around one million people on the Atlantic Coast. In Nova Scotia, efforts have been dedicated to improving stroke treatment outcomes since 2005 [22]. However, further improvements across the province were still needed to meet Canadian best practice guidelines for the treatment of AIS. Specifically, prior to the start of this project, the median DTN for alteplase treatment was nearly double the benchmark of 30 min [6] and patients outside the main city, Halifax, had poorer access to endovascular treatment. Nova Scotia has 10 designated stroke hospitals, and all suspected stroke patients are taken to one of these hospitals within 12 h of onset by paramedics. If the paramedics arrive within 12 h of the symptoms’ onset, they transport the patient to a designated stroke center. However, if they arrive after 12 h, the patient is taken to the closest hospital, which may not be a designated stroke center. The mQIC was led and coordinated by the principal investigator for the ACTEAST project, which was primarily funded by the Canadian Institutes for Health Research (CIHR) Project Grant Scheme. It is important to note that while the project team used ACTEAST to refer to the entire project across all Atlantic provinces, for the interviewees, ACTEAST was understood as being equivalent to what we refer to as the mQIC.

This study adhered to the COREQ (Consolidated Criteria for Reporting Qualitative Research) guidelines to ensure transparency and comprehensive reporting of the qualitative research process. The COREQ checklist, which provides detailed information on various aspects of the study, including participant selection, data collection, and analysis, is available as Appendix A.

Figure 1 illustrates the mQIC in Nova Scotia under the ACTEAST initiative. Out of the 11 hospitals in the region, 10 participated, with the exception being one of the smallest rural stroke centers. Two additional teams joined the mQIC: a team representing an additional urban hospital that receives many stroke patients arriving by private vehicle and provides alteplase treatment, and a team to represent the provincial ambulance service, Emergency Health Services (EHS) Nova Scotia. In total, 11 hospitals in addition to EHS participated in the mQIC.

### 2.2. Setting and Participants

In total, 98 healthcare professionals from Nova Scotia participated in the mQIC; 73 fully consented to the research study. The reason why some of the healthcare professionals did not consent was that they were only marginally participating in the improvements, and they did not have time to commit fully to this project. The first learning session was held on 9 November 2020 in a hybrid fashion (in-person and virtual). All remaining sessions, including the second learning session (held on 8 January 2021), webinars, and site visits with each participating team, were held virtually. All virtual interactions were delivered using Zoom (Zoom Video Communications, v. 5.11.0, San Jose, CA, USA). The reason for moving all aspects to Zoom was due to the COVID-19 pandemic.

Participating teams were supported through the action period by the principal investigator through virtual site visits and webinars. Six webinars were conducted from 24 November 2020 to 29 April 2021 as part of the Nova Scotia Improvement Collaborative, each serving as a platform for knowledge sharing, distribution, and networking among participants. The webinars included a 40 min presentation followed by 20 min for questions and discussion and held via Zoom. All webinars were recorded, with links to the recordings distributed to all participants. All participants were invited to attend the webinars. The topics covered in these webinars were diverse and highly relevant to stroke care, including “Evidence and Best Practice Recommendations for Alteplase and EVT Treatment, and National Institutes of Health Stroke Scale (NIHSS)”, “Case Reviews of Transfers for EVT”, “Skepticism Around tPA and Emergency Consent”, “Changes at a Nova Scotia Hospital” (the specific hospital’s name has been suppressed), and “Endovascular Thrombectomy”. 

Virtual site visits were conducted with all teams via Zoom between 19 February 2021, and 26 April 2021, with one visit scheduled per site towards the end of the mQIC. During the planning of these visits, participants were encouraged to invite additional colleagues from their sites to join. The virtual site visits included an overview of the ACTEAST project, an update from the site on their progress and a review of material covered in the learning sessions for those at the site who were unable to attend. Physician leaders from the Comprehensive Stroke Center in Halifax, NS, were invited to attend the site visits to provide clinical leadership. 

### 2.3. Data Collection

Implementation of the mQIC was evaluated using a validated framework called the Consolidated Framework for Implementation Research (CFIR) for the implementation of best practices. CFIR is an implementation science framework that integrates several theories of implementation research [20] and outlines key factors that can influence implementation. The CFIR comprises 39 constructs grouped under five domains, including innovation characteristics, outer setting characteristics, inner setting characteristics, individual characteristics, and process characteristics. CFIR was used to better understand the barriers and facilitators for implementing acute stroke treatment best practices using an mQIC in Nova Scotia and to help us evaluate the ability of the mQIC to implement best practices for stroke treatment [23].

Semi-structured interviews were conducted with participants from the ACTEAST’s Nova Scotia mQIC after completion of the 6-month improvement collaborative. The evaluation team created and pre-tested a CFIR-based [24] semi-structured interview guide of 35 main questions. The interview guide was pre-tested by a medical staff member who was previously involved in the Quality Improvement and Clinical Research (QuICR) Alberta Stroke Program door-to-needle initiative [16], a similar initiative conducted between 2015 and 2017 in Alberta, Canada. Based on the findings of pre-testing and the users’ feedback, the evaluation team edited the interview guide for language and the questions’ sequencing. The interview guide can be found in Appendix A. The interview guide was sent to all interviewees a couple of days before the interview date as part of a reminder email.

The interviews were conducted from June 2021 to October 2021. Participants were recruited from the mQIC cohort. An initial email invitation was sent to all mQIC participants, inviting them to participate in the interview. Interviews were conducted with those who volunteered in response to the email. Additionally, targeted outreach was performed to selected individuals who either played a significant role in the project or represented a category of participants that was underrepresented among the volunteers. This dual approach to recruitment, namely general invitation and targeted outreach, ensured a diverse and comprehensive set of participants, thereby enriching the quality and scope of the data collected.

For the data collection phase, all interviews were conducted by one team member (S.A.), except the first interview, which two team members (S.A. and K.J.M.) conducted to ensure the interview guide matched the team’s and users’ expectations. At that time, S.A. was a PhD research engineer, while K.J.M. was in the final stages of completing her PhD. As part of the initiative, S.A. participated in several ACTEAST planning meetings for investigators and collaborators during the project’s time. These virtual meetings included some of the interviewees. Accordingly, those participants who attended these meetings may have known his role in the project. K.J.M. also attended some meetings and thus might have been known to the participants.

### 2.4. Data Analysis

Due to the constraints imposed by the COVID-19 pandemic, the interviews were conducted virtually, as it was the only viable option at the time. S.A. conducted the interviews from his home office, while the participants joined from various locations, predominantly their own offices. Only the researcher(s) and the interviewee(s) were present in the interviews. Recorded interviews were conducted through Zoom and transcribed verbatim. Field notes were taken by S.A. during all interviews.

Framework Analysis was employed as our analytical approach, aligning well with the CFIR framework. This method allows for a systematic yet flexible approach to qualitative data analysis, which corresponds with CFIR’s comprehensive and multi-level understanding of implementation contexts [25]. Framework analysis involves a structured process of data management and interpretation, including familiarization with the data; identifying a thematic framework; indexing, charting, and mapping; and interpretation [26]. This systematic approach mirrors CFIR’s structured method of categorizing and analyzing multiple implementation constructs across various domains [20]. The use of framework Analysis guided by CFIR enabled us to apply pre-existing constructs while remaining open to emerging themes, thus capturing both anticipated and unforeseen factors influencing implementation [27]. This method allowed for a nuanced examination of our data, facilitating the identification of key barriers and facilitators to implementation within the CFIR’s comprehensive framework.

Transcripts were cleaned, de-identified, and coded in a line-by-line fashion in NVivo12 (Lumivero, v. 12, Melbourne, Australia) using a pre-defined, deductive CFIR framework codebook (Table 1) [24]. Each coded segment was first identified as either a facilitator or a barrier to implementation. Coding went through several iterations by all three authors (S.A., K.J.M., and N.K.) to reach a consensus within the qualitative evaluation team. N.K. is an Associate Professor and the principal investigator of the ACTEAST project. It should be noted that the transcripts were not returned to the participants for verification, given that they were already burdened by the demands of the COVID-19 pandemic. In addition, the participants were not consulted for feedback on the findings for the same reason. No repeat interviews were needed.

### 2.5. Rigor Considerations

To support the rigor and validity of our findings, we used a methodological approach that included iterative coding and team discussions, with a CFIR framework codebook guiding the coding process. The main author (S.A.) focused solely on the qualitative evaluation of the mQIC, handling tasks from recruiting interviewees to conducting the analysis, separate from the mQIC’s design and implementation. This distinction ensured a rigor or trustworthy evaluation of the outcomes. N.K. implemented the mQIC intervention but she did not participate in data collection or analysis for this study. Similarly, K.J.M. had no involvement with the mQIC intervention or its components. This separation of roles provided a clear demarcation between the mQIC intervention and the qualitative study, minimizing the risk of bias.

## 3. Results

Overall, there were 14 interview participants recruited (14.3% of total participants and 19.2% of consenting participants) from ten entities (nine hospitals and EHS). Detailed information about the 98 participants of the mQIC can be found in [21]. The interviewees’ professions were distributed as follows: four (29% of the interviewees) were stroke coordinators, four (29% of the interviewees) were physicians, three (21% of the interviewees) worked as diagnostic imaging technologists and managers, and three (21% of the interviewees) were other professions such as paramedics and site-level administrators; 64% of the interviewees were women. The participants’ site’s size representation was 36% rural hospitals (typically smaller hospitals located in remote or sparsely populated areas, providing basic healthcare services to the local community, which may have limited resources and specialized services compared with larger hospitals), 36% community hospitals (mid-sized hospitals that serve a wider geographical area than rural hospitals, which offer a broader range of services, including some specialized care, but may still refer complex cases to tertiary hospitals), 14% large tertiary care hospitals (the largest and most specialized hospitals, often located in major urban centers, which offer a comprehensive range of services, including advanced diagnostics, complex surgeries, and specialized treatments, and often serve as referral centers for patients with complex or rare conditions), and 14% reported no association with a specific hospital (e.g., paramedics and provincial-level administrators). While site-level administrators were responsible for coordinating the mQIC within their hospitals, facilitating communication among departments, and ensuring that new protocols were integrated into daily practice, province-level administrators provided strategic oversight, supporting the initiative across multiple hospitals. The interviews varied in length from 22 min to 52 min (mean = 35.99, SD = 9.57).

After line-by-line data capture of barrier–facilitator references for each interview, we identified 458 references and coded these by their CFIR domains. Table 2 summarizes the results of the qualitative analysis regarding the frequencies of barriers and facilitators influencing implementation. In the table, the term “general” refers to citations that are associated with the broader, higher-level construct (e.g., implementation climate) but do not specifically align with any of the more granular constructs (e.g., tension for change, compatibility, etc.). These citations capture aspects of the higher-level construct in its entirety rather than being confined to its individual dimensions. The values presented at higher levels in this table represent aggregated totals that include the sum of the corresponding lower-level numbers in addition to numbers that do not fit in any lower level, which are referred to as “general”. The analysis of the interview references shows that the reported facilitators far outweighed the barriers (384 facilitators to 74 barriers). The interviews revealed a generally positive disposition of participants toward the mQIC and the intervention, on the basis of specific aspects such as the effectiveness of the collaborative approach in improving communication among healthcare teams and the value of the structured learning sessions and action periods in fostering professional development and knowledge sharing. Participants expressed a particular appreciation for how the mQIC facilitated better interdisciplinary collaboration and a more patient-centered approach in acute stroke treatment. No particular differences were detected in the reported barriers or facilitators on the basis of professional background, discipline, hospital size, or location.

As we will expand upon below, in the domain of inner settings (D3), the most frequently cited facilitators were networks and communications (n = 51, D3.2) and culture (n = 36, D3.3). Concerning networks and communications, the participants reported effective dissemination of essential information regarding the various activities within the mQIC. Additionally, the interdisciplinary nature of the teams was deemed a critical factor of success by the interviewees. As for the culture aspect, the respondents highlighted the mQIC’s role in fostering a culture conducive to embracing change at their respective sites, as well as the significance of maintaining an adaptive organizational culture for the successful implementation of the mQIC.

Within the process domain (D5), the most commonly reported facilitator was the engagement of key stakeholders (n = 37, D5.2.5). Besides emphasizing the value of involving stakeholders typically associated with such collaboratives, the interviewees underscored the mQIC’s ability to successfully engage under-represented stakeholders, including emergency staff. This inclusive engagement was considered a vital contributor to the mQIC’s success. In the context of outer setting characteristics (D2), the respondents predominantly identified peer pressure (n = 26, D2.3) as a facilitative factor, noting that it fostered a positive competitive atmosphere within the mQIC.

In the domain of innovation characteristics (D1), design quality and packaging (n = 26, D1.7) emerged as the most frequently cited facilitator. Specifically, the respondents praised the mQIC’s adaptability in response to the constraints imposed by the COVID-19 pandemic on face-to-face interactions, and its successful transition to accommodate virtual activities. Among the individual characteristics (D4), the respondents emphasized the individual stage of change (n = 9, D4.3) as a key domain. The interviewees noted the mQIC’s effectiveness in enhancing various skills, including knowledge acquisition, team cohesion, collaboration, interprofessional communication, and negotiation, leading to improved DTN, informally, according to the interviewees.

On the other hand, most high-frequency barriers were concentrated in the inner settings and process domains. The top of the list showed three constructs of the inner settings domain: resource availability, relative priority, and networks and communications. The two main resources needed by the participants were time and funding. In addition, the mQIC took place during the global COVID-19 pandemic, which led to an increase in many side-tracking responsibilities for many participants. The most frequently coded construct was available resources (n = 19), followed by relative priority (n = 17), respectively. Networks and communications came next, with nine references, followed by key stakeholders (staff) (n = 6) and culture (n = 4). Finally, design quality and packaging, reflecting and evaluating, complexity, relative advantage, leadership engagement, and executing each had three references or fewer. Figure 2 shows the barriers’ frequencies compared with the facilitators’ frequencies for the same constructs. In the following, the key findings are presented by domain. In the following, we will analyse the constructs with te highest frequency using the most significant citations. Additional supporting citations can be found in Appendix A.

### 3.1. (D1) Innovation Characteristics

Three significant constructs of innovation characteristics were identified as facilitators. It was essential for the participants to recognize the strength of the evidence, considering it as a key motivation to participate in the mQIC. This resulted in comprehending the relatively high degree of advantage of the project, which, in turn, led to assigning a high level of priority (inner settings domain) to the related tasks, even amidst the pressing obligations arising from the global COVID-19 pandemic. The interviewees acknowledged the design quality and packaging as another facilitative factor, particularly its adaptability in accordance with the COVID-19 regulations. 

#### 3.1.1. (D1.2) Strength and Quality of the Evidence

The evidence presented at the learning sessions revealed barriers and facilitators to implementing best practices. Despite the mQIC designers’ aim to present the quality and strength of the evidence, some particular categories of participants, such as those in emergency medicine, were not that convinced of the evidence.

On the other hand, many other interviewees showed their appreciation of including the presentation of the evidence, as well as the way it was presented as part of the initiative. Three participants felt their sites were aware of the evidence, but ACTEAST had helped review it and improve understanding of the rationale for the team’s work and the priority of re-evaluating the processes. Interviewee 006 mentioned that “So I think understanding the evidence and understanding the improvement’s potential for the patients made a difference. And again, really, I think it made the biggest impact for the members of our group, who are the ED group, to get on board with a really rapid response in getting the specialties involved, getting radiology involved, getting everyone involved earlier”. Presenting the evidence showed the urgent nature of stroke cases and resulted in moving from a “haphazard” protocol to a “true protocol”, according to Interviewee 009. For Interviewee 012, presenting the evidence improved their knowledge of the stroke treatment. 

The understanding of the evidence changed for five interviewees, as some of the information was presented in a way that was easy to directly relate to improved patient outcomes. Interviewee 007 said “Putting the human behind it instead of just the numbers did give it a different lens”. The understanding of the evidence fluctuated for Interviewee 010 who said that “it was quite amazing to hear how different interpretation of the evidence is from a neurology perspective versus emergency medicine perspective”. 

The impact of the strength and quality of the evidence primarily influenced the implementation of the mQIC rather than direct stroke care. Skepticism among emergency medicine practitioners regarding tPA’s efficacy emerged as a significant barrier, affecting the activation of stroke protocols and overall buy-in for the mQIC. Conversely, positive reception and improved understanding of the evidence among other healthcare professionals acted as a facilitator, leading to enhanced engagement and adherence to structured response protocols. Additionally, the diversity in interpretations of the evidence between neurology and emergency medicine highlighted the challenge of aligning different medical specialties, impacting the cohesive implementation of the mQIC.

#### 3.1.2. (D1.3) Relative Advantage

Because of the chaotic nature of emergency environments, the staff’s understanding of the direct advantages of changes is required to be compelled to implement them. Interviewee 012 said: “once you explain it to them, they usually get it, but it’s just that change piece, that change management piece takes a little bit of time”. 

One advantage of ACTEAST mentioned by an interviewee was that it had started many conversations. For three interviewees, the improvement team and the principal investigator discussed specific improvements that made the process more efficient and improved the DTN and stroke care. For two interviewees, ACTEAST provided a framework for launching initiatives planned before the project started. ACTEAST was also recognized to have enabled more communication between sites and fostered broader discussions around the benefits and risks of lytics versus EVT. In addition, the advantage of monitoring and analyzing data to determine where improvements must occur was acknowledged.

Our analysis of the ACTEAST initiative showed that it mainly impacted the implementation process, not directly improving stroke care. It emphasized the need for effective change management and facilitated important discussions, aiding in the adoption of better practices in emergency settings. Key impacts included improved inter-site communication, deeper treatment discussions, and use of strategic data for ongoing improvement.

#### 3.1.3. (D1.7) Design Quality and Packaging

Two interviewees preferred in-person activities. For example, Interviewee 003 expressed that the virtual format of many ACTEAST activities negatively influenced the project, as it limited networking opportunities, and virtual activities were not healthy, as they involved a lot of sitting and screen time. Interviewee 003 said “and it’s not always the formal part of the day, but it’s the informal side conversations. It’s the who you sit with at lunch and who you strike up a conversation with you. You don’t get that when you’re meeting online”. Nevertheless, the virtual format was the only possible option for meeting under the COVID-19 pandemic restrictions, and it was even considered a positive outcome of the pandemic, allowing gathering many stakeholders efficiently while saving expenses according to four interviewees. Interviewee 001 said “I know at first whenever the decision was we’re going virtual. I kind of thought I don’t know how that’s going to work, but really, I think it worked pretty good”. Four interviewees appreciated the design quality of the project, as all the necessary information was provided with check-ins from the principal investigator. 

The design and packaging of the ACTEAST activities mainly influenced its implementation, with the shift to a virtual format due to COVID-19 eliciting mixed responses. Key aspects such as the provision of comprehensive information and regular check-ins by the principal investigator were vital for engagement and smooth execution of the mQIC during the pandemic, despite not directly impacting stroke care.

### 3.2. (D2) OutersSetting 

Outer settings were generally regarded as facilitators, both in terms of the project’s cosmopolitanism across the entire province of Nova Scotia and the constructive competitive atmosphere among the various locations created by this provincial-level collaboration. 

#### 3.2.1. (D2.1) Needs and Resources of Those Served by the Organization 

Most of the references for this construct were related to improving the stroke process, specifically obtaining the patient’s and/or family’s consent before starting treatment. For nine interviewees, they felt it was essential to get the patient’s or family’s consent for the stroke treatment. Two interviewees stated that calling 911 or presenting to a hospital was an indirect form of consent.

Five interviewees pointed to the emergency and time-sensitive nature of a stroke case, and that consent to stroke treatment should be handled the same way as a trauma case or a heart attack. Interviewee 003 thought that it was unfair to ask the patient or their families to make a decision: “I think it’s more of a process of informing the families or the patient that this is what we would recommend. And unless you object to that, this is what we’re going to do, but I don’t think it’s fair to ask them who are in an absolute state of crisis that you know for their opinion really”. 

Interviewee 004 was more concerned about managing expectations: “You know there’s a lot of them have seen the TV shows, even the commercials in Canadian about stroke, you know? Good thing you called us early. We’ve cured you well; that’s not reality, and so managing those expectations happens through consent”.

Interviewee 005 talked about a patient who was “very upset that someone would take the time to stop and ask him if and give him their information to see if he wanted it. He said, why wouldn’t I? Why would you stop and ask somebody that, when my mind wasn’t processing properly?” The stroke coordinator had to explain that the treatment comes with risks and the patient had to be informed. 

The predominant focus among interviewees on the issue of obtaining the patient’s or family’s consent before starting stroke treatment underlines a critical aspect of patient care. The perspectives varied, with some likening the emergency nature of stroke treatment to trauma or heart attack cases, where implicit consent is assumed. This approach was contrasted by the concerns raised about managing patients’ expectations and the ethical complexities of informed consent in crisis situations. These insights underscore the delicate balance in stroke care between the urgency of treatment and respecting patients’ autonomy.

#### 3.2.2. (D2.2) Cosmopolitanism 

Being a provincial project gave ACTEAST a level of importance in Nova Scotia, according to seven interviewees. For example, Interviewee 001 said “I think because it was provincial project, it kind of gave some added value to it and gave it a level of importance”. Even though, at the province level, there was always a stroke network among the coordinators, ACTEAST was an opportunity for older participants to get to meet and know newly joined ones. Three interviewees highlighted ACTEASTs “invaluable” networking opportunities. According to Interviewee 002, “[ACTEAST] connected local players with each other and then connected those same players with other players across the province”. This includes local specialists that would be collaborators on some future cases. 

The ACTEAST initiative, with its provincial reach, significantly enhanced the mQIC’s implementation in Nova Scotia. Its cosmopolitan aspect boosted the project’s value and regional engagement. The initiative also established a key networking platform, promoting collaboration and knowledge exchange among healthcare professionals. This network facilitated the integration of new participants into the stroke care network, reinforcing a collaborative, province-wide approach to stroke treatment in Nova Scotia.

#### 3.2.3. (D2.3) Peer Pressure

Four interviewees stated that seeing changes in other sites created positive competition. This allowed sites to learn from the experiences of other sites, both positive and negative. Interviewee 001 said, “it always takes the question, how come they can do it and we can’t? So it drives people a little bit”. Interviewee 004 also said, “I’m always a friend of competition”. Seeing other sites’ performance data and changes was positive for nine interviewees. 

According to Interviewee 004, knowing what was going on at the different sites was essential for the tertiary site in terms of realizing that the referring sites were doing what was required to prepare patients before being transferred. 

Other interviewees did not feel that peer pressure played a role in implementing improvements. An interviewee pointed out that peer pressure did not have an impact because the site was already highly motivated. Interviewee 006 highlighted that the overload resulting from the COVID-19 pandemic has diminished the positive effects of peer pressure. Another interviewee pointed out that sites in their particular region do not like to be compared with other places, and it was tricky to consider peer pressure as positive motivation. However, the sites that had an impact from peer pressure were in the same geographical zone with similar resources and sizes. 

The concept of “peer pressure” in the ACTEAST project played a notable role in the implementation of the mQIC, as indicated by the participants’ feedback. Positive competition, as observed by four interviewees, encouraged sites to adopt successful practices from others, enhancing motivation and fostering a culture of continuous improvement. However, not all viewed peer pressure positively; concerns ranged from pandemic-related overload to regional differences in receptiveness to comparative performance. 

### 3.3. (D3) Inner Settings 

The inner settings domain encompassed both the most frequent facilitators and barriers. The availability of resources, including time and funds, was identified as the most frequent barrier. The difficulty of assigning high priority to mQIC activities was another barrier, but this same construct was also seen as a frequent facilitator, indicating that sites were able to prioritize the project. 

#### 3.3.1. (D3.2) Networks and Communications 

The factor of networks and communications was the most frequently mentioned ACTEAST facilitator and one of the most mentioned barriers. Sources of information, according to all interviewees, about stroke-related evidence and the project included webinars and emails among the team members and from the site leadership, the ED, the stroke coordinators, and the principal investigator. Just as one example, Interviewee 002 said, “[The principal investigator] was good to communicate. Our teams were fairly good to communicate as well. So we would have regular meetings with minutes and follow up action items”. 

Challenges to networks and communication were mentioned by three interviewees. These included identifying the team at the beginning of the project, the disjointed teamwork at times according, and significant team changes within the first couple of months 

According to Interviewee 013, the distribution of action over different departments and specialties resulted in slow progress, since the relationship with some departments was challenging. On the other hand, a principal part of ACTEAST was the development of interdisciplinary improvement teams and the facilitation of collaboration within the team. These were well appreciated by four interviewees. 

Teams worked together well, according to 11 interviewees. For example, Interviewee 003 said, “I would dream about them forever. They were gold”. Two interviewees attributed the excellent teamwork to the stroke coordinators and the long collaboration history before ACTEAST, as mentioned by Interviewee 008 who said, “I’m fortune enough that I’ve been around long enough to know the majority of the stakeholders in emergency department”. 

The “networks and communications” element was pivotal in the implementation of the mQIC within the ACTEAST project. Effective communication channels, such as webinars and emails, facilitated the dissemination of information, as noted by all interviewees. However, challenges such as team formation, disjointed teamwork during COVID-19, and turnover posed significant barriers. Despite these, the development of interdisciplinary teams and collaboration was highly valued, contributing to the project’s success.

#### 3.3.2. (D3.3) Culture

The relationship between the sites’ cultures and ACTEAST was bidirectional. On the one hand, the open culture at some sites helped accept the changes introduced by ACTEAST. On the other hand, ACTEAST helped, at some sites, in making the culture more open to changes. The “culture” of healthcare sites significantly influenced the implementation of the mQIC in the ACTEAST project. The interviewees spoke about a culture that resisted change. Three interviewees expressed the difficulty of making changes and resistance to change. Interviewee 002 said, “Any suggestion or changes [conflicted] initially with [what we thought we can do], so it takes a while to break down that barrier, [which] is a characteristic of our culture here”. 

ACTEAST helped the culture to change over time at one site, and it became very open. Interviewee 001 said that the culture “has probably changed a little bit over time [to become] open and willing to try new things”. Interviewee 009 explained how most of the younger and nursing staff were open to change, while doctors were not. 

Adopting changes was welcomed by the teams and leadership at five sites. For example, Interviewee 007 pointed to how the change in the leadership led to a more open culture. However, even with everybody on board, changes still needed a lot of collaboration within sites and accordingly dealing with each sites’ own cultures, according to another interviewee. 

However, to EHS, despite some challenges, change was welcome. Interviewee 007 stated, “prehospital care is an ever-changing environment, so we are always willing to change, but don’t love change”. 

#### 3.3.3. (D3.4) Implementation Climate

The “implementation climate” within different sites played a crucial role in the implementation of the mQIC as part of the ACTEAST project. Eight sites did not prioritize stroke management, often due to their site’s culture and due to COVID-19. However, three sites maintained stroke as the top priority, employing strategies such as dedicated staff and leadership collaboration. On the other hand, feedback was provided through virtual site visits, which were generally positive, as they allowed discussion of site-specific issues and boosted team morale, though some participants struggled to engage or did not find them impactful. 

##### (D3.4.3) Relative Priority 

At eight sites, stroke was not a priority. For example, Interviewee 003 said, “it’s still not seen as important as acute MI or trauma”. Additionally, according to Interviewee 013, the leadership welcomed changes if they did not disturb other processes or require funding. Interviewee 013 said, “if our process is going to cause delays in another part of the system, they want no part of it. If it’s going to cause an excessive expenditure of money, they don’t want any part of it”. 

At other sites, the individual role-related tasks of some participants were prioritized over some ACTEAST activities. COVID-19 and the different waves of the pandemic were taking the highest priority at five sites. One interviewee thought that the pandemic’s priorities were even used as an excuse for not achieving improvements in reducing DTN. The extra load resulting from COVID-19 led Interviewee 006 to a feeling of failure in dealing with competing priorities. 

According to four interviewees, dedicating staff to managing the activities of ACTEAST was the way to keep it prioritized. At three sites, stroke always had the highest priority. Ways mentioned by interviewees to address competing priorities included face-to-face conversations and following these up, dedicated staff, regular meetings, updating senior leadership, collaborating with other sites at the leadership level, and negotiation. For one interviewee, dealing with competing priorities needed openness about the pressures and the barriers, and trying one’s best to work around them. 

The relative priority of stroke management across various sites notably affected the implementation of the mQIC in the ACTEAST project. While some sites de-emphasized stroke care due to competing healthcare needs or resource constraints, others, particularly during COVID-19, faced challenges in maintaining a focus on stroke-related initiatives. In contrast, dedicated staffing and consistent prioritization helped keep stroke care as a primary focus, demonstrating varied approaches to managing competing priorities in healthcare settings.

##### (D3.4.5) Goals and Feedback 

Feedback was provided to the sites participating in the mQIC via a virtual site visit. One interviewee explained how the virtual visit allowed for a discussion of project-related topics with a site outsider. Interviewee 004 said “maybe get some critique from somebody from the outside [principal investigator]”. Interviewee 006 explained how it boosted the team’s morale, generated ideas, and engaged new participants and noticed that the site’s specific struggles and solutions were discussed during the visit. However, for Interviewee 001, the virtual visit was the most challenging part of the project to engage people with, as it was hard to understand its benefit. Accordingly, this interviewee reported that the virtual visits were not impactful.

The provision of feedback via virtual site visits played a crucial role in the implementation of the mQIC within the ACTEAST project. These visits facilitated valuable external critique and insights, and were instrumental in boosting team morale and generating tailored solutions to site-specific challenges. 

#### 3.3.4. (D3.5) Readiness for Implementation 

Leadership engagement was critical for implementation, with some interviewees highlighting the importance of support from higher positions. Other interviewees emphasized a collective leadership approach, as was the case with Interviewee 004, who said, “I mean, they just want us to do what you need to do. Let us know if you need some help so there’s no issues anywhere along the way”. 

Time was a scarce resource, often due to competing commitments such as COVID-19. Interviewee 002 said, “there were times when I felt like I was asking a lot of other people, so I would just do it myself and then it was a lot for myself”. Solutions included task prioritization and efficient planning. 

##### (D3.5.1) Leadership Engagement 

The interviewees felt that the leadership’s engagement had a significant impact on their results. For Interviewee 013, the mQIC did not result in a significant impact because the team could not engage people in higher positions with authority to implement the planned changes. Conversely, one interviewee mentioned that prior roles and good relationships with their leadership resulted in buy-in from the leadership and everyone down. On the other hand, Interviewee 013 felt that leadership was provided by all employees: “We are the leadership. I must say we don’t spend a lot of time talking to people who work at levels above us just because they don’t have the power to make to effect change”. In addition, Interviewee 014 explained that there was no need for processes that go through several levels of authority in a small hospital.

Leadership engagement emerged as a pivotal element in the implementation of the mQIC in the ACTEAST project. The ability to engage higher-level authorities significantly impacted the implementation’s outcomes. Furthermore, a few interviewees highlighted a more grassroots approach to leadership, with decisions made locally and swiftly, especially in smaller hospital settings.

##### (D3.5.2) Available Resources 

Time was the most limited resource for six interviewees. Accordingly, Interviewee 002 mentioned that actions had been carried out by a small group of participants: “you get folks volunteering for teams and then they can’t really dedicate as much time as it required so couple people end up doing more than others; that becomes taxing”. In addition, participants backed out, as per Interviewee 005’s quote mentioned above. One interviewee expressed that some participants, physicians in particular, already had busy schedules. Time was limited because some participants had volunteered for many projects or the extra COVID-19 meetings. 

To work around the busy schedule, prioritizing tasks and a task-oriented operation with specified deadlines were adopted, as per Interviewee 004. The same interviewee recognized that the excellent planning and organization of the mQIC allowed for a smoother time commitment. Other mentions of limited resources by three interviewees included limitations of financial resources, particularly for a stroke coordinator position and other required material. Time constraints significantly impacted the implementation of the mQIC. Limited time availability, exacerbated by the COVID-19 pandemic and other commitments, often resulted in a few participants handling the majority of the workload. Effective strategies, such as prioritizing tasks and setting clear deadlines, were essential for managing these time-related challenges. Financial limitations for critical roles and resources also posed challenges.

##### (D3.5.3) Access to Knowledge and Information 

The information and access to knowledge provided by the mQIC was found to be both beneficial and challenging. For Interviewee 012, education of the operational team was a challenge in implementing changes, as they need to understand the rationale. Interviewee 012 said, “the biggest hurdle is getting everybody educated–not that they’re not on board or not that they don’t understand, because we usually do a pretty good job of front ending that information, so it’s getting them to change, actually, physically change their practice”. According to Interviewee 009, the information and knowledge acquired through the project resulted in tuning protocols to the sites’ particular needs. 

The webinars conducted during the mQIC were a source of knowledge and information. According to six interviewees, the topics covered in the webinars were excellent, even with varied attendance because experts were available in the webinars for comments and questions, even when they were not presenting. According to Interviewee 011, the webinars helped realize the positive effect of speeding up the processes. An interviewee from a presenting site expressed that even when the webinars were not locally beneficial as a learning opportunity for the site, they still increased the overall efficiency of the process. Interviewee 006 explained how webinars that presented successful outcomes for patients transferred for EVT had rejuvenated physicians to make the decision to transfer. 

Two interviewees explained that the topics and conflicting meetings presented challenges to attending the webinars. Interviewee 002 suggested having external experts presenting, as opposed to local ones, as local initiatives can sometimes be limited in their progress. 

Effective dissemination of knowledge was key for implementation. Challenges included educating teams about changes in practices and issues such as inconsistent attendance at beneficial webinars, which faced scheduling conflicts. However, the value of tailoring protocols to site-specific needs was highlighted.

### 3.4. (D4) Individual Characteristics 

Most of the references to individual characteristics were facilitators and related to the stage of change of the participants. These references mainly pertained to the new knowledge or skills acquired through the mQIC.

#### (D4.3) Individuals’ Stage of Change 

The interviews revealed that participants of the mQIC were at different stages of implementing the improvements. According to six interviewees, ACTEAST helped build up and improve skills in terms of “knowledge base and understanding what’s actually happening” (Interviewee 001), “bring[ing] teams together” (Interviewee 002 and Interviewee 004), “collaboration” (Interviewee 006), “interprofessional communication” (Interviewee 008), and the ability to “advocate for the patients” (Interviewee 009). For five interviewees, no skills related to acute stroke treatment were built on or improved. Interviewee 014 attributed this to the excellent stroke training as a radiologist at one of the participating hospitals.

In the mQIC, the participants’ progress varied across different stages, affecting the overall implementation of the project. While some interviewees credited ACTEAST with skill enhancement that was vital for improvements in stroke treatment, others perceived no new skill development.

### 3.5. (D5) Process 

Engaging key stakeholders was a frequently identified facilitator. In general, stakeholders or champions recognized the benefits and importance of the mQIC and actively participated or identified other stakeholders to involve. 

#### 3.5.1. (D5.1) Planning 

The interviews revealed that the action planning provided during the learning sessions of the mQIC was integral to planning improvements. Interviewee 001 said “I really liked having that time. As opposed to, you know, here’s your learning session and you know, maybe you guys can get together at some point later on. I think this way, we were kind of not forced into doing it, but it’s built into your meeting”. Interviewees thought the action planning was impactful even though it was not perfect because of COVID-19, according to Interviewee 005, who said “not that it was perfect either because again, same caveats as previously with COVID really throwing a wrench in some things”. For Interviewee 009, it helped focus efforts and provided a framework and a list of actions. Interviewee 007 explained how having several departments at the table with the right stakeholders involved allowed for better streamlining of the process. At Interviewee 013’s site, different communication and coordination issues resulted in inefficient action plans, as the interviewee explained “They came up with things that needed to be fixed. Then for some things, they determined actions that should be taken. Didn’t really assign them to people and they didn’t really set a date when they should be done”. 

Interviews revealed that the structured planning during the mQIC’s learning sessions was key to its implementation, especially amidst the challenges of COVID-19. The participants appreciated the clear framework and goal-setting, along with the advantages of involving multiple departments in streamlining the process. However, some coordination issues impacting the efficiency of planning were also identified.

#### 3.5.2. (D5.2) Engaging 

Five interviewees found engaging certain departments challenging, while having influential participants and involving emergency staff and EHS physicians were seen as key to success. 

##### (D5.2.5) Key Stakeholders (Staff) 

For five interviewees, engaging particular departments was challenging because of attitudes or competing priorities. For example, Interviewee 002 said, “getting [the] Diagnostic Imaging (DI) [department] engaged was tough. We ended up getting a CT tech, but we couldn’t get a radiologist involved”. Interviewee 003 said, “the nurses in the emergency department are not engaged, and I’m not sure if that’s related to the physician’s attitude or they’re just overworked!” and Interviewee 012 said “physician engagement is tough here, and it’s not that they don’t agree. It’s not that they don’t understand the data; obviously, they do […], [in a small hospital], physicians’ time is very precious and it’s difficult to get somebody committed to an improvement process like this”. 

According to four interviewees, appropriate participants were able to influence their departments. Interviewee 001 mentioned that the mQIC had created an environment where everyone felt efficient. For two interviewees, engaging emergency staff and EHS physicians was a key to success. 

Stakeholder engagement, particularly among staff from various departments, was essential for the mQIC’s implementation. Challenges in engaging certain departments due to competing priorities or attitudes were noted. However, having influential staff who could effectively communicate within their departments significantly aided the initiative’s success.

#### 3.5.3. (D5.4) Reflecting and Evaluating

Reflection on and evaluation of the mQIC showed mixed feelings about the mQIC. For Interviewee 005, everything related to the ACTEAST mQIC was stressful due to the added daily work. 

Eight interviewees expressed that all or most parts of the mQIC and the resulting changes would stay after the end of the mQIC because of its positive outcomes. For example, Interviewee 013 said, “Programs of research like ACTEAST are vital to make sure that a systems change can actually take place”. For Interviewee 013, the mQIC was an ongoing process, as there were always things to fix and improve. The interviewee said, “I’m not sure that the collaboration is ended. It’s an ongoing process, right? Because there are always things to fix and to improve upon”. Interviewee 013 was a stroke neurologist, and all the ACTEAST mQIC’s activities and tasks were part of their daily job. Still, for that same interviewee, sustaining the changes and operationalizing the work was challenging. Interviewee 013 reported that further follow-up after the mQIC ended and continuous feedback data would be excellent.

### 3.6. Other Aspects

While not mainly part of the CFIR framework, answers from the interviewees highlighted aspects such as sustainability, scalability, and other aspects. This study showed that the interviewees believed in the sustainability and scalability of the mQIC, as defined by Klaic et al. [23]. While the evaluation of sustainability in [23] required evaluating the outcomes over extended periods of time, our interview guide included questions on what aspects of the changes would remain after the end of the mQIC and what participants perceived would be challenging to keep up. Most interviewees answered that most of the changes would remain, indicating that the intervention may be sustainable. On the other hand, other features that impact the scalability of an mQIC were also highlighted by the interviewees. Regarding acceptability, only one interviewee mentioned dropout during the mQIC, indicating a minimal dropout rate. In addition, the overall perception of the interviewees of the mQIC was positive. Fidelity can be demonstrated by positive comments related to the excellent planning (delivered as intended). Most interviewees expressed the project’s practicality and ease of delivery, which indicated its feasibility.

### 3.7. Post-Implementation Model

Figure 3 depicts a post-study logic model of the mQIC, integrating key findings from the initiative’s evaluation. Inputs included healthcare professionals’ expertise and resources for conducting virtual learning sessions, which were crucial during the COVID-19 pandemic. Activities such as learning sessions, action planning, and webinars aimed to improve stroke treatment protocols. The outputs reflected enhanced knowledge and shared practices among participants. The outcomes targeted improved treatment processes and reduced treatment times. The model incorporated external influences such as the pandemic, which necessitated virtual engagement and influenced the effectiveness of the activities. The facilitators highlighted in the model included effective communication and stakeholder engagement, while the barriers involved time and funding constraints. This post-study logic model thus succinctly encapsulates the interplay among the initiative’s elements, facilitators, and barriers, providing a clear framework for understanding the mQIC’s impact on improvements in stroke care.

In Figure 3, the construct of networks and communications appears in the categories of both barriers and facilitators. This dual placement reflects the nuanced feedback from the interviewees. While the construct was generally appreciated as a key facilitator provided by the mQIC, there were specific occasions when communication posed challenges. These challenges, although not pervasive throughout the entire mQIC, were significant during particular periods, such as the team’s formation or due to team turnover. It is important to highlight these “occasional” challenges, as they were frequently mentioned in the interviews, despite being limited to specific periods of the mQIC. Similarly, other constructs such as “culture” also appear as both barriers and facilitators in Figure 3, reflecting their high frequency of being mentioned in both positive and negative contexts.

## 4. Discussion

Through this study, we identified various facilitators and barriers for improving the utilization and efficiency of AIS treatment in Nova Scotia through an mQIC. Ultimately, the goal of identifying the barriers and facilitators was to investigate the perceived impact and future implementability of the mQIC in different sites and/or settings. To answer these research questions, semi-structured interviews, containing qualitative questions, were conducted with 14 mQIC participants and analyzed via CFIR.

The barriers most often reported by the participants related to the available resources, namely the lack of available time and financial resources to undertake the implementation. While financial obstacles require more effort and time to overcome, some participants handled the time limitations with planning and prioritizing. This provides important insight into the future use of an mQIC to improve acute stroke treatment, as providing participating teams with strategies and support for planning and prioritization (i.e., project management) can mitigate the time constraints that are ever present in healthcare settings; however, support from the health system and hospital to protect time while participating in quality improvement initiatives should not be overlooked. This is consistent with studies pointing out that extra resources are required to carry out quality improvement [28,29,30] or explicitly point to limited resources as barriers [15,31]. Of note was the fact that COVID-19 was not considered a competing priority by interviewees at sites where stroke was already prioritized. However, for those whose roles were not devoted to stroke treatment and management and were thus involved in COVID-19-related tasks, implementation reportedly caused a massive increase in workload, leading to feelings of failure in managing competing priorities. 

Even though barriers related to networks and communications were among the most frequent barriers, this construct was also the most frequently mentioned facilitator. In some sites, this construct was an occasional barrier; for example, during the team’s creation phase or because of team changes during the mQIC. The teams’ interdisciplinarity [31] was considered by many interviewees to be a facilitator. Still, some signaled the difficulty of communication among different departments. While the team’s formation and dynamics were, at times, complex, a finding also raised in previous work [15,29], the collaboration within the interdisciplinary teams was much appreciated by our study participants. This shows the importance of creating an interdisciplinary team involving individuals from all areas involved in care during the first hours of the stroke (paramedics, ED nurses, emergency physicians, radiologists, and CT technologists) to guide improvements. This critical first step, while challenging, was critical to implementing best practices in our study. The formation of an interdisciplinary team was identified as a critical pre-determinant to successful implementation [32]. Having an interdisciplinary team in place was a pre-condition for effective implementation of the mQIC in Nova Scotia. The team’s diverse skill set and collaborative approach significantly facilitated the implementation process, setting the stage for the success of the mQIC from the outset.

Engaging key stakeholders has previously been identified as a major barrier [29]. For ACTEAST, the physicians were the most impactful stakeholders to engage; their points of view about the treatment influenced the level of their departments’ engagement. Also, the leadership’s engagement has been identified in other work as a facilitator [28,31]; this was also a factor appreciated by our study participants during ACTEAST. Communication was not only considered a facilitator at the site level, but the project’s cosmopolitanism was also considered a key facilitator, providing communication and sharing of experiences among different sites [15]. 

Some interviewees identified a culture of initial resistance to change as a barrier. Initial resistance to change can be defined as an initial reluctance or opposition that individuals or organizations could exhibit when faced with changes or new initiatives. The ACTEAST mQIC was reported to have a positive culture change. The mQIC’s design quality was appreciated, and the provincial-level collaboration was reported to frequently create a positive, friendly competitive environment. The interviewees reported that the changes implemented during the mQIC would be sustained. Most interviewees expressed their satisfaction with the design quality, including the virtual format of activities, which was the only possible way to meet at the time due to COVID-19 restrictions on gathering. Still, a few participants noted the lack of networking opportunities and the unhealthiness of extended screen time. The webinars were felt to be the least beneficial aspects of the mQIC, followed by the site visits. In future, the number of webinars could potentially be limited to a few critical topics. Additionally, further consideration should be given to the site visits, as they can be time-consuming for both the mQIC’s organizers and the sites. If used, the site visits should be organized in a mutually beneficial manner, while allowing the opportunity to connect with each site. Similar thoughts on-site visits during the ACTEAST mQIC were found in other studies; however, they found that having the participants play a more active role in the site visits helped enhance their utility [33]. 

The mQIC created some peer pressure to improve their acute stroke treatment processes. Most interviewees agreed on the positive impact of peer pressure created by implementation efforts. Only a few sites had reservations. The use of peer pressure in healthcare is not new, and several studies have found it to influence improvements [12,34,35,36,37]. 

The results identified “relative priority” and “available resources” as major challenges, highlighting the need for focused strategies on resource allocation and prioritization of QIC within the organization. The cyclical nature of these challenges, where low priority could lead to reduced resources, further lowering priority, was noted. This interplay between prioritization and resource allocation aligns with the findings in [38] on the importance of systematic resource prioritization.

It was also observed that some groupings of constructs were mentioned by multiple interviewees, indicating recurring themes across different perspectives. For example, the combination of “readiness for implementation” and “engaging key stakeholders” was frequently discussed together, highlighting a common challenge in balancing these two aspects. Similarly, “available resources” and “engaging key stakeholders” were often linked, suggesting that the effectiveness of communication networks directly influences the overall environment for implementation. Also, “networks and communications” and “implementation climate”, as well as “culture” and “implementation climate” were often mentioned by the same interviewees. In the following paragraphs, we will be looking at these combinations of constructs. 

In addition, other barriers mentioned related to ”readiness for implementation” and “engaging key stakeholders”. It is clear that an organization’s readiness for implementation could be directly influenced by the level of engagement by key stakeholders. A lack of stakeholder engagement could signify a lack of organizational readiness and vice versa. This observation aligns with the findings from [39], where organizational readiness was linked to stakeholder engagement.

Similarly, barriers related to “available resources” and “engaging key stakeholders” were also prevalent. The availability of resources could be a significant factor in the ability to engage key stakeholders. Insufficient resources may lead to inadequate engagement activities, which, in turn, could affect the overall success of the mQIC. This aligns with the findings from [40] that suggested that effective stakeholder engagement relies on clear objectives and identification of the necessary resources.

Other barriers were noted in “networks and communications” and “implementation climate”. Ineffective communication networks within an organization can create an unfavorable implementation climate, as poor connections and information gaps among team members can cause misunderstandings about an mQIC’s objectives, impacting the implementation climate. This is supported by findings in [41] that linked poor communication to decreased performance and productivity, affecting both the work environment and the implementation climate. In addition, barriers related to “relative priority” and “networks and communications” were found. The relative priority given to the mQIC within the organization could be influenced by the effectiveness of internal communications. If the mQIC is not effectively communicated as a priority, it may not receive the attention or resources it needs for successful implementation. According to [42], communication has a major role in employees’ engagement and morale during programs of change, which can affect how initiatives such as the ACTAEST mQIC are prioritized.

Also, barriers related to “culture” and “implementation climate” were cited by Interviewees 002 and 012. The organizational culture could have a direct impact on the implementation climate. A culture that is resistant to change or innovation may create a climate that is not conducive to implementing new initiatives such as the mQIC. In [43], the organizational culture, especially in terms of flexibility and adaptability, can significantly influence the implementation climate, affecting levels of productivity.

## 5. Challenges and Future Prospects

Employing the CFIR framework, we coded the interview data into domains and constructs, facilitating our processes of coding and synthesis. Our thematic synthesis, based on CFIR, highlighted how these constructs influenced broader themes such as organizational dynamics in implementation. This approach provided a detailed view of the various factors affecting the mQIC’s implementation, encompassing both internal and external influences. The interview questions were framed in a positive manner to explore the perceived impact and effectiveness of the mQIC on acute stroke treatment. Additionally, this study recognized limitations in the recruitment and data saturation. The limited number of interview participants may be attributed to the constraints imposed by the COVID-19 pandemic, which affected the availability and willingness of potential participants to engage in interviews. In addition, in this study, we prioritized analyzing the distribution of citations across the CFIR constructs over the traditional goal of reaching data saturation. Our focus was on understanding how different factors influenced the implementation of the mQIC, and mapping the distribution of these factors provided critical insights into the patterns of barriers and facilitators encountered by the participants. This approach was justified by recent critiques of saturation in qualitative research, which argued that saturation may not always be necessary or feasible, particularly when the goal of research is to explore the breadth of data rather than to generate new theoretical insights [44,45]. Instead, the concept of “information power,” as proposed by [46], suggests that the adequacy of the sample size should be determined by the richness and relevance of the data collected, relative to the study’s aims. Given our objective to distribute and categorize citations across CFIR references, the emphasis on distribution rather than saturation allowed us to capture a comprehensive overview of the dynamics of implementation without being constrained by the need to achieve a point of “no new information”. This approach also aligned with practical considerations, such as the limited resources and the exploratory nature of the study, where capturing a wide range of perspectives was more valuable than exhaustively covering every possible viewpoint. While our approach in these aspects was aligned with our research objectives, it may have influenced the nature and analysis of the responses. Future studies will aim to overcome these limitations. Also, this publication represents the first step in analyzing the implementation of the mQIC within Nova Scotia. As a preliminary study, the focus was on understanding the specific context and dynamics within this province. Consequently, the generalizability of the results to other regions or broader populations has not been fully explored in this article. However, we recognize the importance of this aspect and plan to address it in a forthcoming publication. The upcoming study will extend the analysis to include all Atlantic provinces, allowing for a more comprehensive evaluation of the generalizability of the findings presented here. This future work will carefully examine how the results from Nova Scotia apply to other provinces, providing a more robust and generalizable understanding of the mQIC’s implementation across the Atlantic region.

The reliance on virtual interviews, necessitated by the COVID-19 pandemic, may have inadvertently excluded non-tech-savvy participants, potentially skewing the data towards those more comfortable with technology. This limitation suggests that future studies should consider incorporating alternative methods, such as phone interviews or in-person sessions where possible, to ensure broader participant inclusion.

Moreover, it is crucial to assess whether the implementation of the mQIC led to tangible improvements in outcomes, such as the process’s efficiency, time metrics, and increments in the ratio of patients receiving treatment, as these are critical indicators of the initiative’s success. As the quantitative assessment is still in progress, it will later provide an opportunity to validate and compare our qualitative findings. At this stage, however, we relied on the insights gathered from interviewees to understand the perceived impact and challenges of the mQIC.

## 6. Conclusions

In this study, we described the use of the CFIR to evaluate the implementation of an mQIC aiming to improve ischemic stroke treatment in Nova Scotia, Canada. In our study, interviewees expressed that the mQIC was effective in implementing improvements to acute stroke treatment processes. Using the virtual format as a facilitator eliminated the need for travel and reduced time commitments, thereby facilitating greater participation. However, the lack of available time was a barrier to prioritizing the work, which can be mitigated by providing strategies to assist teams in prioritization and planning. The study showed the importance of creating an interdisciplinary improvement team, which may be critical when improving acute stroke treatment processes. Additionally, the potential benefits of peer pressure among sites can also be key to acute stroke improvement efforts. 

The study also revealed ways that the mQIC could be improved to be even more beneficial. Webinars should only be conducted in areas of keen interest to the participants; quality over quantity should be central when conducting webinars. Additionally, site visits should be planned in a manner that makes the visits more beneficial to the site participants. However, most aspects of the mQIC were found to be beneficial. 

## Figures and Tables

**Figure 1 healthcare-12-01801-f001:**
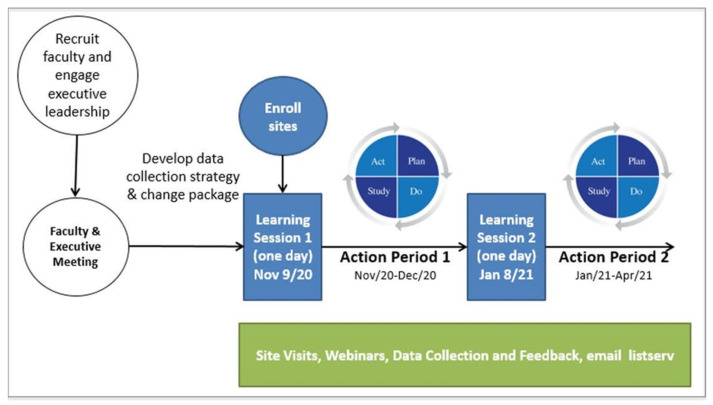
The improvement collaborative’s process for the ACTEAST project (from [21]).

**Figure 2 healthcare-12-01801-f002:**
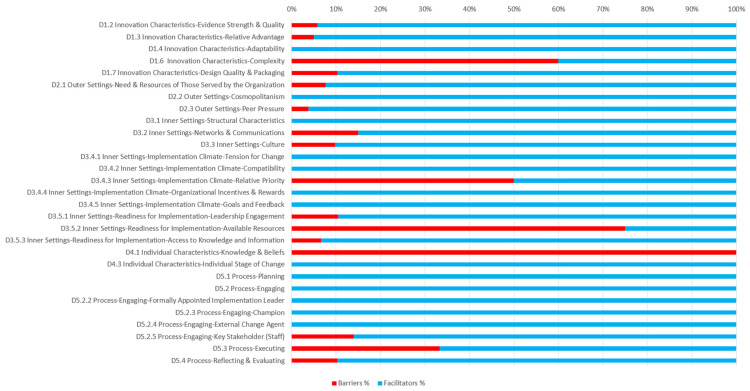
Barriers’ frequencies compared with facilitators’ frequencies for the same constructs.

**Figure 3 healthcare-12-01801-f003:**
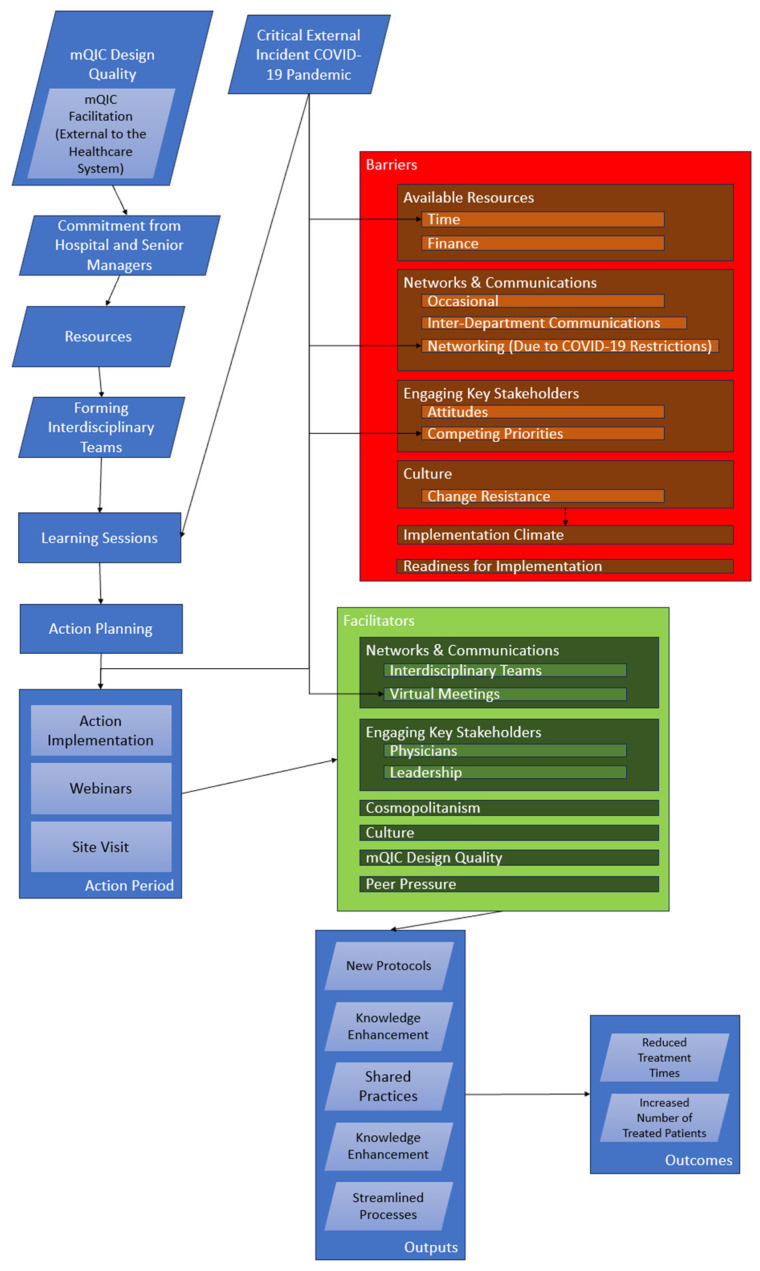
A post-study logic model of the mQIC.

**Table 1 healthcare-12-01801-t001:** The CFIR constructs and their definitions.

Domain/Construct Code	Description
Domain 1: Innovation characteristics	Characteristics of the intervention
D1.1 Source of innovation	Perception of the intervention’s origin (external or internal)
D1.2 Strength and quality of the evidence	Perception of the quality and validity of evidence for the intervention
D1.3 Relative advantage	Perception of the intervention’s advantage over alternative solutions
D1.4 Adaptability	Degree of adaptability of the intervention to local needs
D1.5 Trialability	Ability to test and potentially reverse the intervention
D1.6 Complexity	Perceived difficulty of implementing the intervention
D1.7 Design quality and packaging	Perceived excellence in the presentation of the intervention
D1.8 Cost	Costs of the intervention and its implementation
Domain 2: Outer setting characteristics	The economic, political, and social context within which an organization resides
D2.1 Patients’ needs and resources	The extent to which patients’ needs and barriers are known and prioritized
D2.2 Cosmopolitanism	The degree to which an organization is networked externally
D2.3 Peer pressure	Pressure to implement an intervention due to competition or norms
D2.4 External policy and incentives	Influence of external strategies, policy, regulations, and incentives
Domain 3: Inner setting characteristics	Structural, political, and cultural contexts of the implementation process
D3.1 Structural characteristics	Features of an organization such as size, maturity, and social architecture
D3.2 Networks and communications	Quality of social networks and communications within an organization
D3.3 Culture	Norms, values, and basic assumptions of a given organization
D3.4 Implementation climate	Organizational capacity for change and receptivity to an intervention
D3.4.1 Tension for change	Perception of the current situation as intolerable or needing change
D3.4.2 Compatibility	Alignment of the intervention with individual norms and existing workflows
D3.4.3 Relative priority	Perception of the importance of the implementation within the organization
D3.4.4 Incentives and rewards	Use of extrinsic and intangible incentives to promote the intervention
D3.4.5 Goals and feedback	Clear communication and alignment of the goals with staff feedback
D3.4.6 Learning climate	Climate promoting leader–team collaboration, safety, and reflective thinking
D3.5 Readiness for implementation	Indicators of organizational commitment to implementation
D3.5.1 Leadership engagement	Leaders’ dedication and accountability in implementation of the innovation
D3.5.2 Available resources	Resources allocated for the implementation and operations
D3.5.3 Access to knowledge and information	Ease of acquiring and comprehending information for integration of the task
Domain 4: Individual characteristics	Individuals engaged in the intervention or implementation process
D4.1 Knowledge and beliefs	Individuals’ attitudes toward and understanding of the intervention
D4.2 Self-efficacy	Individuals’ belief in their ability to achieve the implementation’s goals
D4.3 Individuals’ stage of change	Phase an individual is in towards skilled and sustained use of the intervention
D4.4 Individuals’ identification with the organization	Individuals’ perception of and commitment to the organization
D4.5 Other personal attributes	Personal traits, such as motivation and learning style, influencing implementation
Domain 5: Process characteristics	Coordinated actions for effective, flexible use of the intervention
D5.1 Planning	Quality of planning for implementation of the intervention
D5.2 Engaging	Strategies to involve individuals in the implementation of the intervention
D5.2.1 Opinion leaders	Influential individuals shaping peers’ attitudes towards the intervention
D5.2.2 Formally appointed implementation leader	Formally designated individuals responsible for implementing the intervention
D5.2.3 Champion	Individuals actively driving the intervention amidst organizational resistance
D5.2.4 External change agent	External associates influencing or steering the innovation’s decisions positively
D5.2.5 Key stakeholders (staff)	Staff directly impacted by the innovation
D5.2.6 Participants in the innovation (patients)	The organization’s clientele participating in the innovation
D5.3 Executing	Carrying out or accomplishing the implementation according to plan
D5.4 Reflecting and evaluating	Evaluative feedback and regular debriefing on the progress of implementation

**Table 2 healthcare-12-01801-t002:** Frequency of reported barriers and facilitators comments coded by CFIR domains/constructs.

Domain/Construct	Barriers	Facilitators
Domain 1: Innovation characteristics	9	64
D1.2 Strength and quality of evidence	1	16
D1.3 Relative advantage	2	19
D1.4 Adaptability	0	1
D1.6 Complexity	3	2
D1.7 Design quality and packaging	3	26
Domain 2: Outer settings	2	54
D2.1 Need and resources of those served by the organization	1	12
D2.2 Cosmopolitanism	0	16
D2.3 peer pressure	1	26
Domain 3: Inner settings	51	170
D3.1 Structural characteristics	0	1
D3.2 Networks and communications	9	51
D3.3 Culture	4	36
D3.4 Implementation climate	16 + 1(general)	44 + 1(general)
D3.4.1 Tension for change	0	10
D3.4.2 Compatibility	1	0
D3.4.3 Relative priority	15	16
D3.4.4 Organizational incentives and rewards	0	4
D3.4.5 Goals and feedback	0	14
D3.5 Readiness for implementation	21	37
D3.5.1 Leadership engagement	2	17
D3.5.2 Available resources	18	6
D3.5.3 Access to knowledge and information	1	14
Domain 4: Individual characteristics	1	9
D4.1 Knowledge and beliefs	1	0
D4.3 Individuals’ stage of change	0	9
Domain 5: Process	11	87
D5.1 Planning	0	10
D5.2 Engaging	6	45 + 2(general)
D5.2.2 Formally appointed implementation leader	0	1
D5.2.3 Champion	0	4
D5.2.4 External change agent	0	3
D5.2.5 Key stakeholder (staff)	6	37
D5.3 Executing	2	4
D5.4 Reflecting and evaluating	3	26

## Data Availability

For data access, please contact Shadi Aljendi (shadi.aljendi@unb.ca).

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
