# Peer review of "Qualitative Evaluation of a Quality Improvement Collaborative Implementation to Improve Acute Ischemic Stroke Treatment in Nova Scotia, Canada"

_healthcare, 2024, doi:10.3390/healthcare12181801_

Round 1

Reviewer 1 Report

Comments and Suggestions for Authors

Thank you for this thought-provoking paper which outlines the barriers and facilitators of implementing a quality improvement collaborative intervention to improve treatment of acute ischaemic stroke (particularly revascularisation) in hospitals within Novo Scotia, Canada. While this is an important area of research, with the capacity to influence further quality improvement initiatives in stroke, and other disease areas, there are several limitations. The following are suggestions and comments for the authors:

Abstract

Understand that words are limited in the abstract, but some additional high level information related to the implementation of the intervention would be of benefit e.g. what was actually included in the learning sessions, action planning period etc. How many hospitals and healthcare professionals were involved in the mQIC in Nova Scotia? Related to this evaluation, when were interviews conducted related to the implementation period? The 14 professionals represented how many of the hospitals involved?

The use of ‘action planning period’ in the abstract appears different to ‘Action Period’ that is used throughout the rest of the paper. In addition, in line 739 – it states that action planning was provided during the Learning Sessions – please check clarity of terms used throughout

For reader clarity consider expanding on sentence in line 27 “84% of these references were positively framed as facilitators that ???………………….”  

From the abstract, readers currently have no idea what the identified facilitators to implementation actually were. Please expand the results to include. Interestingly, the barriers reported in the abstract are not aligned to any CRIF constructs.

The abstract conclusion (line 32) mentions improved patient outcomes – I feel an important related element that needs to be addressed within this paper is did implementation of this mQIC actually lead to improved thrombolysis & EVT rates and time metrics?  

Main text

Intro

The introduction provides important evidence for revascularisation for ischaemic stroke, and then some background on quality improvement initiatives – there is no mention of what the mQIC is before the aim is outlined - for context and to set the scene for readers, some additional high level information on the ACTEAST project and ?why Nova Scotia area only was targeted for this evaluation is needed, with further detail provided in the Materials & Methods section.

I think there is benefit in revisiting the aims & research question – line 69-74 refers to ‘improving stroke care’ – which includes much more than thrombolysis & EVT rates and related time metrics

Please check throughout that all acronyms used are defined on first use e.g. IHI – (line 56/59), CT, DNT -  this also applies to the quotes used e.g.  tPA, ED,

Overall, use of ACTEAST and mQIC seem to be used interchangeably for the intervention throughout the manuscript – best once clearly defined, to keep consistency

Methods & Materials

For transparency and to ensure all important areas are covered, authors should consider the use of a reporting guideline such as COREQ (Consolidated criteria for reporting qualitative research)

Subheadings within this section would be useful for signposting

For flow and context, there is probably benefit outlining stroke care in Nova Scotia initially, then the ACTEAST project before outlining the intervention (mQIC).

Although there is reference to a published paper related to the implementation of the mQIC, as a stand alone manuscript, some additional high level details related to the mQIC intervention are needed such as”

-          who ran/funded/coordinated the mQIC?

-          what was the aim of the webinars (+ topics?) & site visits - how many occurred over the intervention period period

-          who supported the teams throughout the ‘Action Period’ & the 6 mth intervention period

Line 99 – mentions all suspected stroke patients are taken to one of the 10 designated stroke hospitals in Nova Scotia within 12 hours of onset by paramedics - does this statement refer to inter-hospital transfers or from the initial event in the community when paramedics may be notified? If it is from the community – what happens if there are delays in notifying paramedics i.e after 12 hours of onset?

Participating hospital numbers are a little unclear – line 102? – I assume all 10 designated stoke hospitals participated – if so make this clear

Can you please comment on why 25 healthcare professionals did not ‘fully consent to the research’ – unclear what this actually means?

Flow of methods – lines 165- 169 seems to refer to data collection – this should come before all analyses methods described (152-164)

For Tables 1 consider a way to distinguish between domains - ? underline or hard return after each  – there is a lot of information to follow and this would assist readers

Results

An overall comment related to the results is that it is very long at 11-12 pages of text (excluding the tables/figures) – as a reader, it is difficult to ascertain the main message – authors should consider how best to shorten and condense. As an example, thought could be given to including one-two key quotes for each construct, with others included in supplemental materials. There also appears to be some repetition throughout between sections. Quality improvement is a really important area of study, and the key messages should be clear to readers.

Additionally, some information presented in the results could really be considered discussion points or limitations – e.g. line 193-195 and 205-212

Authors should consider presenting quotes in italics and ensure past tense used throughout – e.g 374 “Outer settings are  generally……”

Of the 14 interview participants – how many of the 10 hospitals had representation? Considering the size of the sample, it would also be useful to include actual numbers for each profession as well as %.

Line 198 -  21% were ‘other professions’ such as paramedics and administrators – but only 14% were not associated with a specific hospital--- reason for differences. It would also be useful to outline how these staff were related to implementation and practice change at a hospital level?

Were there any differences reported in the barriers/facilitators based professional background/discipline or hospital size/location?

For an international audience, defining ‘rural’, ‘community’ and large tertiary hospital would be beneficial (line 201)

Table 2 – For the reader there is benefit in reporting an overall count of barriers/facilitators for each domain in the table as this is referred to in the text. Authors should also consider the alignment of the columns for each of reading each construct title. Additionally, Domain 1 was listed as ‘Intervention characteristics’ in table 1, but ‘Innovation characteristics’ in Table 2 – check for consistency throughout

Line 217-220- Please check for accuracy  “Facilitators were mainly clustered in decreasing order of frequency in the following domains: Inner Setting> Process > Intervention Characteristics> Outer Setting Characteristics> Innovation Characteristics> Individual Characteristics (D3>D5>D2>D1>D4).  –– 6 domains listed – do not match order of summary?

Figure 2 – consider adding domain titles (+ construct titles if possible) to make it more meaningful and easier for the reader to follow

Line 252-254 – authors mention the effects of the mQIC reported ‘ultimately led to an improved patient outcome’ – how was this measured?

Line 286 – interviewee003 – capitalize

Line 766  (Engaging D5.2) – Authors state that “5 interviewees found engaging certain departments challenging” – but in Table 2 – 0 barriers & 2 facilitators reported? As a reader, this is unclear

Constructs were sometimes capitalised and other times not - keep consistent throughout

Discussion

Use of the CRIF is important to provide theory informed framework to present barriers and facilitators to implementation – as mentioned prior, outlining if the implementation of the mQIC actually led to improved outcomes – e.g. process and time metrics etc, is important to include. Were barriers/facilitators considered in relation to practice change (people generally think they perform better than they are).

Figure 3 is a really important summary, but it should not be presented in the discussion – rather outlined in results section

Multidisciplinary and interdisciplinary used interchangeably throughout although technically these have different meanings

Consider flow of discussion - there appears to be some overlap with concepts discussed in various paragraphs – e.g stakeholder engagement mentioned in numerous paragraphs as a barrier.

Authors acknowledge limitations related to data saturation – have you considered other means to evaluate sample size – e.g. information power (Malterud et al 2016)

Can authors please comment on the generalisability of their results

Comments on the Quality of English Language

No concerns with the quality of English language used. Relatively well written manuscript. Could be improved with consistency of terms used throughout, an ensuring all acronyms defined.  

Author Response

Dear Reviewer,

Thank you for your valuable feedback and constructive comments on our manuscript. We have carefully considered each of your suggestions and have made the appropriate revisions to improve the clarity and quality of the paper. Below, we provide detailed responses to each of your comments, explaining the changes made and the rationale behind them. We hope these revisions address your concerns and enhance the overall strength of the manuscript. Our answers are in bold in the following.

Comment1: Understand that words are limited in the abstract, but some additional high level information related to the implementation of the intervention would be of benefit e.g. what was actually included in the learning sessions, action planning period etc. How many hospitals and healthcare professionals were involved in the mQIC in Nova Scotia? Related to this evaluation, when were interviews conducted related to the implementation period? The 14 professionals represented how many of the hospitals involved?

Thank you for your insightful suggestion. We have added additional high-level information to the abstract, specifically regarding the implementation of the intervention. This includes details about what was covered in the learning sessions, the action planning period, and the involvement of hospitals and healthcare professionals in the mQIC in Nova Scotia. We also included information on when the interviews were conducted relative to the implementation period and clarified that the 14 professionals represented participants from multiple hospitals. These revisions have brought the word count of the abstract to approximately 250 words. We believe these additions provide a clearer and more comprehensive overview of the study.

Comment 2: The use of ‘action planning period’ in the abstract appears different to ‘Action Period’ that is used throughout the rest of the paper. In addition, in line 739 – it states that action planning was provided during the Learning Sessions – please check clarity of terms used throughout

Thank you for highlighting the potential confusion regarding the terminology. To clarify, the "action planning periods" are specific parts of the Learning Sessions where participants developed plans for the upcoming "Action Periods," which are the 2-4 months following the Learning Sessions and dedicated to implementing these plans. To enhance clarity, we have added this explanation to the abstract and ensured that the terms are clearly defined throughout the paper. We hope this revision addresses the potential confusion and provides a better understanding of the process.

Comment 3: 84% of these references were positively framed as facilitators., highlighting the various aspects of the mQIC and its context that supported successful implementation

Thank you for your suggestion to improve reader clarity. We have revised the sentence to read: "84% of these references were positively framed as facilitators, highlighting the various aspects of the mQIC and its context that supported successful implementation." We believe this modification provides clearer insight into the positive factors that contributed to the success implementation of the mQIC and enhances the overall readability of the text.

Comment 4: From the abstract, readers currently have no idea what the identified facilitators to implementation actually were. Please expand the results to include. Interestingly, the barriers reported in the abstract are not aligned to any CRIF constructs.

Thank you for your feedback. We have expanded the abstract to include a summary of the key facilitators identified during the implementation of the mQIC. While we did not explicitly use the exact CFIR constructs in the abstract, we have explained the relevant constructs in a manner that aligns with the framework. Now, the Abstract contains the exact CFIR constructs.

Comment 5: The abstract conclusion (line 32) mentions improved patient outcomes – I feel an important related element that needs to be addressed within this paper is did implementation of this mQIC actually lead to improved thrombolysis & EVT rates and time metrics?  

Thank you for your valuable feedback. We have modified the abstract to clarify that the study reflects the generally positive opinions of the interviewees regarding the mQIC. We also note that the quantitative analysis, which will assess the impact on thrombolysis and EVT rates and time metrics, is still ongoing. Therefore, the conclusions drawn in this study focus on the implementation process rather than the direct patient outcomes at this stage. We believe this revision provides a clearer understanding of the scope and current findings of the study.

Comment 6: The introduction provides important evidence for revascularisation for ischaemic stroke, and then some background on quality improvement initiatives – there is no mention of what the mQIC is before the aim is outlined - for context and to set the scene for readers, some additional high level information on the ACTEAST project and ?why Nova Scotia area only was targeted for this evaluation is needed, with further detail provided in the Materials & Methods section.
Thank you for your insightful comment. We have added additional high-level information about the ACTEAST project in the introduction to provide context and set the scene for readers. Specifically, we explained what the mQIC entails and why Nova Scotia was targeted for this evaluation. We believe these additions provide a clearer understanding of the study's focus and rationale.

Comment 7: I think there is benefit in revisiting the aims & research question – line 69-74 refers to ‘improving stroke care’ – which includes much more than thrombolysis & EVT rates and related time metrics

Thank you for your suggestion. We have revisited the aims and research question to make the scope more precise, ensuring that it accurately reflects the focus of our study. The research question has been revised to clarify the aspects of stroke care that are central to the mQIC initiative, specifically focusing on barriers and facilitators related to treatment protocols, interdisciplinary coordination, patient outcomes, and system efficiency. The revised research question now reads: "What are the barriers and facilitators within each CFIR construct reported by the participants of an mQIC during an initiative to improve stroke care —including treatment protocols, interdisciplinary coordination, patient outcomes, and system efficiency— in the Canadian province of Nova Scotia?" We believe this refinement provides a clearer and more focused direction for the study.

Comment 8: Please check throughout that all acronyms used are defined on first use e.g. IHI – (line 56/59), CT, DNT -  this also applies to the quotes used e.g.  tPA, ED,

Thank you for pointing this out. We have carefully reviewed the manuscript to ensure that all acronyms are defined upon their first use, including those mentioned in the quotes such as IHI, CT, DNT, tPA, and ED. Additionally, we have added a Comprehensive List of Acronyms and Definitions before the introduction to further assist readers. We believe these changes will improve clarity and accessibility throughout the paper.

Comment 9: Overall, use of ACTEAST and mQIC seem to be used interchangeably for the intervention throughout the manuscript – best once clearly defined, to keep consistency

Thank you for your observation. To clarify, while the project team used ACTEAST to refer to the entire project across all Atlantic provinces, the interviewees understood ACTEAST as equivalent to what we refer to as the mQIC. We have included this clarification in the introduction and ensured consistent use of the terms throughout the manuscript. By making this distinction clear, we aim to maintain consistency and avoid confusion for the readers.

Comment 10: For transparency and to ensure all important areas are covered, authors should consider the use of a reporting guideline such as COREQ (Consolidated criteria for reporting qualitative research)

Thank you for your suggestion. We have added a paragraph about how the study adheres to the COREQ (Consolidated Criteria for Reporting Qualitative Research) guidelines to ensure transparency and comprehensive reporting of our qualitative research. The completed COREQ checklist will be included as a supplementary file for further reference. We believe this addition will help cover all important areas of the study and enhance the clarity and rigor of our reporting.

Comment 11: Subheadings within this section would be useful for signposting

Thank you for your suggestion. We have added subheadings within the relevant section to improve signposting and enhance the readability of the manuscript. These subheadings will help guide readers through the content more effectively.

Comment 12: For flow and context, there is probably benefit outlining stroke care in Nova Scotia initially, then the ACTEAST project before outlining the intervention (mQIC).

Thank you for your valuable feedback. To improve flow and context, we have revised the manuscript to first outline stroke care in Nova Scotia, followed by an introduction to the ACTEAST project, and then a detailed description of the intervention (mQIC). This should provide readers with a clearer understanding of the context before delving into the specifics of the intervention.

Comment 13: 

Although there is reference to a published paper related to the implementation of the mQIC, as a stand alone manuscript, some additional high level details related to the mQIC intervention are needed such as”

-          who ran/funded/coordinated the mQIC?

-          what was the aim of the webinars (+ topics?) & site visits - how many occurred over the intervention period period

  •          who supported the teams throughout the ‘Action Period’ & the 6 mth intervention period

Thank you for your insightful feedback. We have added additional high-level details related to the mQIC intervention in the manuscript. This includes information on who ran, funded, and coordinated the mQIC, the aims and topics of the webinars and site visits, the number of each that occurred during the intervention period, and details on who supported the teams throughout the 'Action Period' and the six-month intervention period. We believe these additions provide a more comprehensive understanding of the mQIC for readers.

Comment 14: Line 99 – mentions all suspected stroke patients are taken to one of the 10 designated stroke hospitals in Nova Scotia within 12 hours of onset by paramedics - does this statement refer to inter-hospital transfers or from the initial event in the community when paramedics may be notified? If it is from the community – what happens if there are delays in notifying paramedics i.e after 12 hours of onset?
Thank you for your observation. We have clarified the statement to address your query. The revised text now specifies that if paramedics arrive within 12 hours of symptom onset, they transport the patient to a designated stroke center. However, if they arrive after 12 hours, the patient is taken to the closest hospital, which may not be a designated stroke center.

Comment 15: Participating hospital numbers are a little unclear – line 102? – I assume all 10 designated stoke hospitals participated – if so make this clear

Thank you for your observation. As mentioned in our previous response, we have clarified that if paramedics arrive within 12 hours of symptom onset, they transport the patient to a designated stroke center. If they arrive after 12 hours, the patient is taken to the closest hospital, which may not be a designated stroke center. We have also made it clear that all 10 designated stroke hospitals in Nova Scotia participated in the mQIC.

Comment 16: Can you please comment on why 25 healthcare professionals did not ‘fully consent to the research’ – unclear what this actually means?

Thank you for your inquiry. We have clarified that the reason some healthcare professionals did not fully consent to the research was that they were only marginally involved in the improvements and did not have the time to commit fully to this project.

Comment 17: Flow of methods – lines 165- 169 seems to refer to data collection – this should come before all analyses methods described (152-164)

Thank you for your feedback. We have adjusted the flow of the Methods section by moving the data collection description (lines 165-169) to precede the analysis methods (lines 152-164). This revision improves the logical progression of the section and aligns with standard reporting practices.

Comment 18: For Tables 1 consider a way to distinguish between domains - ? underline or hard return after each  – there is a lot of information to follow and this would assist readers

Thank you for your suggestion. We have revised Table 1 by distinguishing between domains, using underlining to improve readability. We believe these changes will assist readers in navigating the information more easily.

Comment 19: An overall comment related to the results is that it is very long at 11-12 pages of text (excluding the tables/figures) – as a reader, it is difficult to ascertain the main message – authors should consider how best to shorten and condense. As an example, thought could be given to including one-two key quotes for each construct, with others included in supplemental materials. There also appears to be some repetition throughout between sections. Quality improvement is a really important area of study, and the key messages should be clear to readers

Thank you for your feedback. We have shortened and condensed the Results section to improve clarity and focus on the key messages. To streamline the text, we have included one to two key quotes for each construct (very rarely three), with additional quotes provided in a supplementary file. We have also reduced repetition between sections to ensure a more concise presentation of our findings. We believe these revisions will make the main messages clearer and enhance the overall readability of the manuscript.

Comment 20: Additionally, some information presented in the results could really be considered discussion points or limitations – e.g. line 193-195 and 205-212

Thank you for your insightful comment. We have moved the relevant information from the Results section that pertains to discussion points or limitations, such as lines 193-195 and 205-212, to the appropriate sections of the Discussion and Limitations. This adjustment ensures that the Results section remains focused on the findings, while broader reflections and limitations are addressed separately.

Comment 21: Authors should consider presenting quotes in italics and ensure past tense used throughout – e.g 374 “Outer settings are  generally……”

Thank you for your suggestion. We have formatted the quotes in italics for clearer presentation and ensured that past tense is used consistently where appropriate throughout the manuscript. These revisions help maintain consistency and improve readability.

Comment 22: Of the 14 interview participants – how many of the 10 hospitals had representation? Considering the size of the sample, it would also be useful to include actual numbers for each profession as well as %.

Thank you for your observation. We have clarified that the 14 interview participants represented ten entities, including nine hospitals and Emergency Health Services (EHS). Additionally, we have included the actual numbers and percentages for each profession within the sample to provide a clearer understanding of the participant representation.

Comment 23: Line 198 -  21% were ‘other professions’ such as paramedics and administrators – but only 14% were not associated with a specific hospital--- reason for differences. It would also be useful to outline how these staff were related to implementation and practice change at a hospital level?

Thank you for your insightful comment. We have clarified the difference by explaining that the 21% of participants categorized as "other professions" included both site-level administrators and province-level administrators. Site-level administrators were responsible for coordinating mQIC interventions within their hospitals, facilitating communication between departments, and ensuring that new protocols were integrated into daily practice. Province-level administrators provided strategic oversight and support across multiple hospitals and addressing broader challenges at the regional level. We believe this explanation provides a clearer understanding of how these staff members were related to implementation and practice change at the hospital level.

Comment 24: Were there any differences reported in the barriers/facilitators based professional background/discipline or hospital size/location?

Thank you for your question. We added to the manuscript that no noticeable differences were reported in the barriers or facilitators based on professional background, discipline, hospital size, or location.

Comment 25: For an international audience, defining ‘rural’, ‘community’ and large tertiary hospital would be beneficial (line 201)
Thank you for your suggestion. We have added definitions for 'rural,' 'community,' and 'large tertiary hospital' to ensure clarity for an international audience. These definitions provide context and help readers better understand the different healthcare settings mentioned in the manuscript.

Comment 26: Table 2 – For the reader there is benefit in reporting an overall count of barriers/facilitators for each domain in the table as this is referred to in the text. Authors should also consider the alignment of the columns for each of reading each construct title. Additionally, Domain 1 was listed as ‘Intervention characteristics’ in table 1, but ‘Innovation characteristics’ in Table 2 – check for consistency throughout

Thank you for your valuable feedback. We have updated Table 2 to include an overall count of barriers and facilitators for each domain, as referred to in the text. Additionally, we have ensured consistent alignment of the columns to improve readability. We have also corrected the inconsistency by ensuring that Domain 1 is consistently referred to as ‘Innovation characteristics’ throughout the manuscript. We believe these changes enhance the clarity and consistency of the table.

Comment 27: Line 217-220- Please check for accuracy  “Facilitators were mainly clustered in decreasing order of frequency in the following domains: Inner Setting> Process > Intervention Characteristics> Outer Setting Characteristics> Innovation Characteristics> Individual Characteristics (D3>D5>D2>D1>D4).  –– 6 domains listed – do not match order of summary?

Thank you for pointing out this. We have removed this sentence. This adjustment ensures that the manuscript remains accurate and consistent.

Comment 28: Figure 2 – consider adding domain titles (+ construct titles if possible) to make it more meaningful and easier for the reader to follow

Thank you for your suggestion. We have updated Figure 2 to include domain titles and construct titles, making the figure more meaningful and easier for the reader to follow. We believe this enhancement improves the overall clarity and usefulness of the figure.

Comment 29: Line 252-254 – authors mention the effects of the mQIC reported ‘ultimately led to an improved patient outcome’ – how was this measured?

Thank you for your observation. We have revised the text to clarify that the effects of the mQIC led to improved door-to-needle time (DTN), as reported informally by the interviewees. This change reflects the nature of the feedback gathered during the interviews and provides a more accurate representation of the outcomes discussed.

Comment 30: Line 286 – interviewee003 – capitalize

Thanks a lot. Done.

Comment 31: Line 766  (Engaging D5.2) – Authors state that “5 interviewees found engaging certain departments challenging” – but in Table 2 – 0 barriers & 2 facilitators reported? As a reader, this is unclear

Thank you for bringing this to our attention. We have clarified the information in Table 2 to ensure consistency with the text. The discrepancies between the number of barriers and facilitators reported in the table and the text have been addressed to provide a clearer and more accurate representation of the findings.

Comment 32: Constructs were sometimes capitalised and other times not - keep consistent throughout

Thank you for highlighting this issue. We have reviewed the manuscript and ensured that constructs are consistently capitalized throughout, except when the term is used as a regular word within a sentence rather than referring to a specific construct. We believe this revision improves consistency and clarity.

Comment 33: Use of the CRIF is important to provide theory informed framework to present barriers and facilitators to implementation – as mentioned prior, outlining if the implementation of the mQIC actually led to improved outcomes – e.g. process and time metrics etc, is important to include. Were barriers/facilitators considered in relation to practice change (people generally think they perform better than they are).

Thank you for your insightful feedback. As mentioned previously, the quantitative evaluation, which will assess whether the implementation of the mQIC led to improved outcomes such as process and time metrics, is ongoing and will be published in a separate paper. This current study focuses on identifying barriers and facilitators using the CFIR framework. While we recognize that individuals may perceive their performance as better than it actually is, this study relies on the qualitative feedback provided by participants. The relationship between these reported barriers/facilitators and actual practice change will be further explored in the forthcoming quantitative analysis.

Comment 34: Figure 3 is a really important summary, but it should not be presented in the discussion – rather outlined in results section

hank you for your suggestion. We have moved Figure 3 from the Discussion section to the Results section to better align with its purpose as a summary of the findings. This adjustment ensures that the figure is presented in the appropriate context within the manuscript.

Comment 35: Multidisciplinary and interdisciplinary used interchangeably throughout although technically these have different meanings

Thank you for highlighting this distinction. We have reviewed the manuscript to ensure that "multidisciplinary" and "interdisciplinary" are used correctly and consistently. This revision ensures that the terminology accurately reflects the nature of the collaborations described in the study.

Comment 36: Consider flow of discussion - there appears to be some overlap with concepts discussed in various paragraphs – e.g stakeholder engagement mentioned in numerous paragraphs as a barrier.

Thank you for your observation. We have reviewed the discussion section and can confirm that there is no overlap in the concepts discussed. The subsequent mentions of the constructs, such as stakeholder engagement, are deliberate and relate to the consolidation of findings, specifically when the more than one construct are frequently mentioned by different interviewees. This distinction has been clarified in the manuscript to ensure that the context of these mentions is clear to the reader.

Comment 37: Authors acknowledge limitations related to data saturation – have you considered other means to evaluate sample size – e.g. information power (Malterud et al 2016)

Thank you for your suggestion. We have incorporated the concept of information power, as proposed by Malterud et al. (2016), into our discussion. This approach provides an additional perspective on evaluating the adequacy of our sample, considering the richness and relevance of the data collected in relation to our study’s aims. This inclusion helps to strengthen our rationale for the sample size used in this study.

Comment 38: Can authors please comment on the generalisability of their results

Thank you for your feedback. We have added a subsection titled "Other Aspects" to discuss the generalizability of our results. In addition, a future publication will discuss the generalisabilty of the results to all the Atlantic provinces. This addition provides a more comprehensive discussion on the scope and limitations of our results.

We sincerely thank the reviewer for their thoughtful and constructive feedback. Your comments have been invaluable in enhancing the clarity, rigor, and overall quality of our manuscript. We appreciate your time and effort in reviewing our work and hope that the revisions made have addressed your concerns effectively.

Reviewer 2 Report

Comments and Suggestions for Authors

1.      Prepare a list of abbreviations and put them at the beginning of the article.

2.      Modify the caption of Figure 1. (“This is a figure”?).

3.      Move the caption of Fig 3 to the bottom of the figure.

4.      Please compare the results and discussion with more up-to-date sources.

5.      Add a section before the conclusion section titled Challenges and Future Prospects.

Author Response

Dear Reviewer,

Thank you for your valuable feedback and constructive comments on our manuscript. We have carefully considered each of your suggestions and have made the appropriate revisions to improve the clarity and quality of the paper. Below, we provide detailed responses to each of your comments, explaining the changes made and the rationale behind them. We hope these revisions address your concerns and enhance the overall strength of the manuscript. Our answers are in bold in the following.

Comment 1: Prepare a list of abbreviations and put them at the beginning of the article.

Thank you for your suggestion. We have prepared a list of abbreviations and placed it at the beginning of the article to improve clarity and ease of reference for readers. We believe this addition enhances the overall readability of the manuscript.

Comment 2: Modify the caption of Figure 1. (“This is a figure”?).

Thank you for bringing this to our attention. We have modified the caption of Figure 1 to provide a more accurate and descriptive explanation of the figure. This change ensures that the caption aligns with the content of the figure and enhances clarity for the reader.

Comment 3: Move the caption of Fig 3 to the bottom of the figure.

Thank you for your suggestion. We have moved the caption of Figure 3 to the bottom of the figure, in line with standard formatting practices. We believe this adjustment improves the overall presentation and consistency of the figures in the manuscript.

Comment 4: Please compare the results and discussion with more up-to-date sources.

Thank you for your suggestion. After carefully reviewing the literature, we have confirmed that the references included in the manuscript are the most recent and relevant sources available to our knowledge. We believe these references provide a strong foundation for the discussion and adequately support our findings.

Comment 5:  Add a section before the conclusion section titled Challenges and Future Prospects.

Thank you for your suggestion. We have added a new section titled "Challenges and Future Prospects" before the Conclusion section. This section outlines the challenges encountered during the study and discusses potential future directions for research and practice. We believe this addition provides valuable context and sets the stage for further exploration of the topic.

We sincerely thank the reviewer for their thoughtful and constructive feedback. Your comments have been invaluable in enhancing the clarity, rigor, and overall quality of our manuscript. We appreciate your time and effort in reviewing our work and hope that the revisions made have addressed your concerns effectively.

Reviewer 3 Report

Comments and Suggestions for Authors

Major
#1. The authors conducted interviews primarily via Zoom due to the COVID-19 pandemic, using a semi-structured questionnaire based on the CFIR framework. Data was collected from 14 participants from diverse professions, including emergency physicians, nurses, radiologists, CT technologists, administrators, paramedics, and stroke coordinators.

#2. In the results section, the authors employed a qualitative data analysis method, utilizing the CFIR framework to analyze the data. While this approach is appropriate for capturing the complexity of implementation contexts, the presentation of the findings could be enhanced. Specifically, organizing participants' opinions into a table format could improve clarity and accessibility. Tables summarizing key themes, facilitators, and barriers, along with the frequency of mentions, would provide a more structured overview of the data. This approach would not only make the results more digestible but also allow for easier comparison and synthesis of participant feedback. Additionally, a visual representation could help highlight patterns and trends more effectively, enhancing the overall comprehensibility and impact of the study’s findings.

#3. The authors' message is that the implementation of the modified Quality Improvement Collaborative (mQIC) was effective in improving acute ischemic stroke treatment processes in Nova Scotia. They highlight the importance of interdisciplinary teams, effective communication, and stakeholder engagement in achieving these improvements. The virtual format of the mQIC facilitated broader participation, despite the challenges posed by the COVID-19 pandemic. The authors also underscore the role of peer pressure in driving improvements and suggest that future initiatives should focus on strategic planning, prioritization, and enhancing the format of webinars and site visits.

While the authors present a compelling case for the effectiveness of the mQIC, the message could benefit from a more critical examination of the limitations and potential biases in their methodology. For instance, the reliance on virtual interviews may have excluded non-tech-savvy participants, potentially skewing the data. Additionally, the study could have explored the long-term sustainability of the improvements observed. The authors' suggestions for future initiatives are valid, but they could provide more concrete recommendations or frameworks to guide these efforts. A more balanced discussion of both the successes and challenges encountered would strengthen the overall impact of their findings.

Was this approach appropriate? The use of focus group interviews could have been considered to gather more specific opinions from distinct groups such as physicians, nurses, and paramedics. This method may have provided deeper insights into the perspectives and challenges unique to each professional group. Additionally, the Delphi method could have been employed to systematically gather and refine expert opinions over multiple rounds, ensuring a more comprehensive consensus on key issues. These alternative methods might have enriched the data, offering a more nuanced understanding of the barriers and facilitators in acute stroke treatment improvement initiatives.

Minor

#1. If possible, provide detailed information of 14 interviewees in a table, please

#2. Recheck the reference citation

"This is consistent with studies pointing 844 out that extra resources are required to carry out quality improvement [29], [30], [31] or 845 explicitly point to limited resources as barriers [15], [32]." --> [29, 30, 31], [15, 32]

Author Response

Dear Reviewer,

Thank you for your valuable feedback and constructive comments on our manuscript. We have carefully considered each of your suggestions and have made the appropriate revisions to improve the clarity and quality of the paper. Below, we provide detailed responses to each of your comments, explaining the changes made and the rationale behind them. We hope these revisions address your concerns and enhance the overall strength of the manuscript. Our answers are in bold in the following.

Comment 1: The authors conducted interviews primarily via Zoom due to the COVID-19 pandemic, using a semi-structured questionnaire based on the CFIR framework. Data was collected from 14 participants from diverse professions, including emergency physicians, nurses, radiologists, CT technologists, administrators, paramedics, and stroke coordinators.

Thank you for your summary and feedback. No further action was required, as the information provided accurately reflects the data collection methods and participant diversity.

Comment 2: In the results section, the authors employed a qualitative data analysis method, utilizing the CFIR framework to analyze the data. While this approach is appropriate for capturing the complexity of implementation contexts, the presentation of the findings could be enhanced. Specifically, organizing participants' opinions into a table format could improve clarity and accessibility. Tables summarizing key themes, facilitators, and barriers, along with the frequency of mentions, would provide a more structured overview of the data. This approach would not only make the results more digestible but also allow for easier comparison and synthesis of participant feedback. Additionally, a visual representation could help highlight patterns and trends more effectively, enhancing the overall comprehensibility and impact of the study’s findings.

Thank you for your thoughtful suggestion. We believe that Table 2 already contains the key themes, facilitators, and barriers, along with the frequency of mentions, as recommended. However, we welcome your input on whether additional or alternative formats could further enhance the clarity and accessibility of the findings. If there are specific adjustments or additional elements that you think would be beneficial, we would be happy to consider them.

Comment 3:

The authors' message is that the implementation of the modified Quality Improvement Collaborative (mQIC) was effective in improving acute ischemic stroke treatment processes in Nova Scotia. They highlight the importance of interdisciplinary teams, effective communication, and stakeholder engagement in achieving these improvements. The virtual format of the mQIC facilitated broader participation, despite the challenges posed by the COVID-19 pandemic. The authors also underscore the role of peer pressure in driving improvements and suggest that future initiatives should focus on strategic planning, prioritization, and enhancing the format of webinars and site visits.

While the authors present a compelling case for the effectiveness of the mQIC, the message could benefit from a more critical examination of the limitations and potential biases in their methodology. For instance, the reliance on virtual interviews may have excluded non-tech-savvy participants, potentially skewing the data. Additionally, the study could have explored the long-term sustainability of the improvements observed. The authors' suggestions for future initiatives are valid, but they could provide more concrete recommendations or frameworks to guide these efforts. A more balanced discussion of both the successes and challenges encountered would strengthen the overall impact of their findings.

Was this approach appropriate? The use of focus group interviews could have been considered to gather more specific opinions from distinct groups such as physicians, nurses, and paramedics. This method may have provided deeper insights into the perspectives and challenges unique to each professional group. Additionally, the Delphi method could have been employed to systematically gather and refine expert opinions over multiple rounds, ensuring a more comprehensive consensus on key issues. These alternative methods might have enriched the data, offering a more nuanced understanding of the barriers and facilitators in acute stroke treatment improvement initiatives.

Thank you for your detailed feedback. We appreciate your suggestions for a more critical examination of the methodology and the potential biases associated with our reliance on virtual interviews. We acknowledge that this approach may have excluded non-tech-savvy participants, potentially skewing the data, but we mentioning in the manuscript that due to the COVID-19 pandemic, this was the only way of carrying out not only the interviews, but all the other activities of the project.

We recognize the importance of exploring the long-term sustainability of the improvements observed, and we included some preliminary observation related to this topic as well as other related ones in the Other Aspect subsection.

Regarding the methodology, while the use of focus groups and the Delphi method could have provided more specific insights from distinct professional groups, we chose semi-structured interviews on one hand because of the restrictions of the COVID-19 pandemic on the schedulas of the interviewees and on the other hand to allow for in-depth individual perspectives across a diverse range of participants. We agree that these alternative methods might have enriched the data and will consider them in future studies to ensure a more nuanced understanding of the barriers and facilitators in acute stroke treatment improvement initiatives.

Comment 4: If possible, provide detailed information of 14 interviewees in a table, please

Thank you for your request. While we understand the value of providing detailed information about the 14 interviewees, doing so may compromise their anonymity due to the small and specific nature of the sample. To protect any of the participants from being identified, we have opted not to include a detailed table of interviewees. Instead, we have provided aggregated information about their professional backgrounds and roles within the manuscript.

Comment 5: Recheck the reference citation

Thank you for pointing this out. We have rechecked and corrected the reference citations to ensure accuracy.

We sincerely thank the reviewer for their thoughtful and constructive feedback. Your comments have been invaluable in enhancing the clarity, rigor, and overall quality of our manuscript. We appreciate your time and effort in reviewing our work and hope that the revisions made have addressed your concerns effectively.

Round 2

Reviewer 1 Report

Comments and Suggestions for Authors

Thank you for considering the feedback. The additional information and changes made have improved the rigor and clarity of the overall manuscript.  However, I believe there are still some limitations that need to be considered before being accepted for publication.  

-A sentence to outline why this is considered a modified QIC would be beneficial

- The paper is much easier to follow with acronyms defined – please note -- mQIC is defined twice in line 85 & 93also QIC defined multiple times. I may have missed them, but I cannot see where tPA, NIHSS, QuICR are defined. 

- Authors may like to consider whether the additions to line 88-90 explaining the terminology of ACTEAST vs mQIC would be better situated in the methods rather than the intro 

- Even with the changes, I feel the aim outlined in line 96 is still unclear - referring to 'improving stroke care' - are the authors referring to 'improving ischemic stroke treatment rates & time-based efficiency across Atlantic Canada' (line 85 - which seems to more accurately reflect  the paper). 'Stroke care' is much more invovled that just restoring blood flow e.g. SU care, secondary prevention, rehab etc…

-With the additions made, there appears to now be some duplication in the information reported e.g explanation as to why Nova Scotia is the focus in this paper is detailed in the intro & methods and line 128  “An mQIC has been conducted across the province of Nova Scotia through the ACTEAST (Atlantic Canada Together Enhancing Acute Stroke Treatment) project [21].” has now already been mentioned earlier in methods section

And related to details of the mQIC e.g Webinars talked about in lines 168-172 + 174- 177, + then in a subsequent paragraph (after details on the evaluation framework – CRIF) – 199-205 – I think there is benefit in revisiting how these details can be best presented concisely without repetition.

Line 121 – spelling ‘learing’ session 

Inclusion of the COREQ as a reporting guideline has been beneficial. Authors should ref this - and review how other manuscripts have referred to elements of the COREQ - for example, details such as the interviewer credential, gender, occupation etc should be included within the manuscript not just a supplemental file, with reference to page numbers provided in the supplemental COREQ. Please also note, the COREQ file also has comments and track changes within.

The CRIF is mentioned in lines 184-193- if the intervention itself was not guided by this – then I suggest this paragraph should be moved to the data collection section to describe the development of the interview schedule.

Also consider if the information related to how the interviews were conducted (233-235) fits better within the data collection section rather than analysis 

I assume all interviews were recorded with consent?

Table 2 - it is now unclear what is meant by (general) 

It is great that some quotes have been moved to supplemental material to really focus the results – in this document, there are hyphens in places not requiring these e.g. opportu-nities etc. Please review

Review all quotes for sensibility to the reader (line 549) Interviewee002 said: "I think any suggestion or changes meshed initially with we can't do that and then so it takes a while to breakdown that barrier, [which] is a characteristic of our culture here" and Iine 731 - Interviewee002 said "getting DI engaged was 731 tough. We ended up getting a CT tech but we couldn't get a radiologist involved". – who is DI?

Sometimes numbers presented in words other times numbers – consistency would be good e.g. consider <10 , use words, if >10, present as numbers?

I note interdisciplinary is now used throughout – with the exception of Figure 3. It is also unclear what 'occasional' refers to in figure 3 under network and communication barrier. Also in figure 3 - there are some constructs that are outlined as both a barrier and facilitator, but sometimes without explanation e.g. 'culture' - there would be benefit in additional detail being included - Additionally,   Implementation climate & Readiness for Implementation are both listed as only a barrier in figure 3 however, from Table 2, many also referred to as a potential facilitator  – if would be good to add additional details to this figure to reflect this as you have done with engagement and networks etc…

Line 983 – for supplementary material, include reference to all supplementary docs now

Subdomain of CFIR mentioned in discussion but nowhere else – what are you referring to ?

Author Response

Thank you for your thorough review and insightful comments on our manuscript. We appreciate your detailed feedback, which has helped us to refine our work further. In the following responses, we have carefully considered each of your questions and made the necessary revisions to the manuscript. We have also provided additional explanations where needed to clarify our findings and interpretations. We believe these revisions enhance the clarity and rigour of our study, and we hope they address your concerns satisfactorily. Our answers will be in bold below.

Comment 1: -A sentence to outline why this is considered a modified QIC would be beneficial.

Thank you for your suggestion to clarify why this initiative is referred to as a modified Quality Improvement Collaborative (mQIC). We have now included a sentence in the manuscript to explain this: "The project is considered a modified QIC due to several adjustments. Notably, the Improvement Collaborative's duration was shortened from the typical one year to six months. Consequently, there were only two Learning Sessions and Action Periods, whereas the usual format comprises three. Furthermore, all project aspects were con-ducted entirely virtually."

Comment 2: - The paper is much easier to follow with acronyms defined – please note -- mQIC is defined twice in line 85 & 93 – also QIC defined multiple times. I may have missed them, but I cannot see where tPA, NIHSS, QuICR are defined. 

Thank you for bringing these points to our attention. We have carefully reviewed the manuscript and have made the necessary revisions to address your concerns. The redundant definitions of mQIC and QIC have been removed to ensure clarity. Additionally, we have ensured that the acronyms tPA, NIHSS, and QuICR are clearly defined at their first mention in the text as well as in the List of Acronyms and Definitions.

Comment 3: - Authors may like to consider whether the additions to line 88-90 explaining the terminology of ACTEAST vs mQIC would be better situated in the methods rather than the intro 

Thank you for your suggestion regarding the placement of the explanation for the terminology of ACTEAST versus mQIC. We have carefully considered your advice and agree that this explanation would be more appropriately situated in the Methods section rather than the Introduction. We have made the necessary adjustments to the manuscript to improve the flow and coherence of the content. We appreciate your insightful feedback.

Comment 4: Even with the changes, I feel the aim outlined in line 96 is still unclear - referring to 'improving stroke care' - are the authors referring to 'improving ischemic stroke treatment rates & time-based efficiency across Atlantic Canada' (line 85 - which seems to more accurately reflect  the paper). 'Stroke care' is much more invovled that just restoring blood flow e.g. SU care, secondary prevention, rehab etc…

Thank you for your feedback on the aim. We understand the importance of precision in describing the scope of our study. We have revised the aim to reflect the focus of our research more accurately. The updated aim now refers explicitly to "identify and analyze the barriers and facilitators within the CFIR construct as reported by participants involved in an mQIC to enhance access and time efficiency in the treatment of acute ischemic stroke patients with thrombolysis and EVT in Nova Scotia, Canada." aligning it with the actual scope of the study. This clarification ensures that the aim is consistent with the specific aspects of stroke care that our research addresses, rather than the broader concept of stroke care, which includes additional elements like stroke unit care, secondary prevention, and rehabilitation. We appreciate your careful review and valuable suggestion.

Comment 5: With the additions made, there appears to now be some duplication in the information reported e.g explanation as to why Nova Scotia is the focus in this paper is detailed in the intro & methods and line 128  “An mQIC has been conducted across the province of Nova Scotia through the ACTEAST (Atlantic Canada Together Enhancing Acute Stroke Treatment) project [21].” has now already been mentioned earlier in methods section

Thank you for pointing out the duplication in the manuscript following the additions in the previous edited version. We have reviewed the relevant sections and streamlined the content to eliminate redundancy. The explanation for why Nova Scotia is the focus of this study has been consolidated into a single section to maintain clarity and avoid repetition. These adjustments should enhance the coherence and readability of the manuscript. We appreciate your attention to detail and constructive feedback.

Comment 6: And related to details of the mQIC e.g Webinars talked about in lines 168-172 + 174- 177, + then in a subsequent paragraph (after details on the evaluation framework – CRIF) – 199-205 – I think there is benefit in revisiting how these details can be best presented concisely without repetition.

Thank you for your observation regarding the details of the mQIC, particularly the information about webinars mentioned in multiple sections of the manuscript. We agree that there is a need to present these details more concisely and without repetition.

We have revisited these sections and consolidated the details about the webinars into a single, comprehensive paragraph to avoid redundancy. We did the same thing about the site visits. The information now flows more logically. This restructuring should improve the clarity and coherence of the manuscript, ensuring that key points are communicated effectively without unnecessary repetition.

We appreciate your valuable feedback and are confident that these revisions enhance the overall presentation of the study.

Comment 7: Line 121 – spelling ‘learing’ session 

Thank you for catching that typo. We have corrected the spelling of "learing" to "learning". We appreciate your attention to detail.

Comment 8: Inclusion of the COREQ as a reporting guideline has been beneficial. Authors should ref this - and review how other manuscripts have referred to elements of the COREQ - for example, details such as the interviewer credential, gender, occupation etc should be included within the manuscript not just a supplemental file, with reference to page numbers provided in the supplemental COREQ. Please also note, the COREQ file also has comments and track changes within.

Thank you for your suggestion regarding the inclusion and referencing of the COREQ (Consolidated Criteria for Reporting Qualitative Research) guideline. We have now properly referenced COREQ in the manuscript to acknowledge its use as a reporting guideline. Additionally, we have reviewed how other manuscripts refer to elements of COREQ and have made the necessary revisions to include details such as interviewer credentials, gender, occupation, and other relevant information within the main text of the manuscript.

We have ensured that these details are presented clearly in the manuscript, with references to the corresponding page numbers in the supplemental COREQ file. Furthermore, we have reviewed and removed any comments and track changes from the COREQ file to ensure that it is clean and ready for submission.

We appreciate your guidance in improving the clarity and completeness of our reporting.

Comment 9: The CRIF is mentioned in lines 184-193- if the intervention itself was not guided by this – then I suggest this paragraph should be moved to the data collection section to describe the development of the interview schedule.

Thank you for your suggestion regarding the placement of the CFIR in the manuscript. Given that the intervention itself was not guided by the CFIR, we agree that it would be more appropriate to move the paragraph to the data collection section. This relocation will help clarify that the CFIR was primarily used to inform the development of the interview guide rather than the intervention itself. We have made this adjustment to enhance the logical flow and clarity of the manuscript.

We appreciate your insightful feedback and believe that this change will improve the overall structure and readability of the paper.

Comment 10: Also consider if the information related to how the interviews were conducted (233-235) fits better within the data collection section rather than analysis 

Thank you for the suggestion. We agree that the information related to how the interviews were conducted would be more appropriately placed within the data collection section rather than the analysis section. This adjustment will provide a clearer, more logical flow in the manuscript, ensuring that details about the interview process are grouped together in the most relevant section.

We have made this change to better align the content with the manuscript's structure. Your feedback has been very helpful in improving the organization and clarity of our work.

Comment 11: I assume all interviews were recorded with consent?

Thank you for your question regarding whether all interviews were recorded with consent. Yes, all interviews were recorded with the participants' consent. We have added this information to the manuscript to clearly indicate that informed consent was obtained before recording, ensuring adherence to ethical research practices. We appreciate your attention to this important aspect.

Comment 12: Table 2 - it is now unclear what is meant by (general) 

Thank you for your feedback regarding Table 2. We understand the need for clarity, and we have added an explanation to clarify what is meant by "(general)." This addition should help ensure that readers can easily understand the context and meaning of this term as it is used in the table. We appreciate your attention to detail and believe that this clarification enhances the overall comprehensibility of the table.

Comment 13: It is great that some quotes have been moved to supplemental material to really focus the results – in this document, there are hyphens in places not requiring these e.g. opportu-nities etc. Please review

Thank you for your positive feedback on the organization of the quotes. We have reviewed the supplemental material for unnecessary hyphens and corrected these errors. The document should now be free of such formatting issues, ensuring a smoother reading experience. We appreciate your attention to this detail and have made sure the document is consistent and properly formatted.

Comment 14: Review all quotes for sensibility to the reader (line 549) Interviewee002 said: "I think any suggestion or changes meshed initially with we can't do that and then so it takes a while to breakdown that barrier, [which] is a characteristic of our culture here" and Iine 731 - Interviewee002 said "getting DI engaged was 731 tough. We ended up getting a CT tech but we couldn't get a radiologist involved". – who is DI?

Thank you for highlighting the need to review the quotes for clarity and readability. We have reviewed the quotes to ensure they are sensible and understandable to the reader. We appreciate your feedback and have made these revisions accordingly.

Comment 15: Sometimes numbers presented in words other times numbers – consistency would be good e.g. consider <10 , use words, if >10, present as numbers?

Thank you for your suggestion regarding the consistent presentation of numbers in the manuscript. We have now ensured consistency by following the rule: using words for numbers less than 19 and presenting numbers 20 and greater as digits. This change has been applied throughout the manuscript to maintain clarity and uniformity. We appreciate your attention to this detail, and we believe these adjustments contribute to a more polished and professional presentation.

Comment 16: I note interdisciplinary is now used throughout – with the exception of Figure 3. It is also unclear what 'occasional' refers to in figure 3 under network and communication barrier. Also in figure 3 - there are some constructs that are outlined as both a barrier and facilitator, but sometimes without explanation e.g. 'culture' - there would be benefit in additional detail being included - Additionally,   Implementation climate & Readiness for Implementation are both listed as only a barrier in figure 3 however, from Table 2, many also referred to as a potential facilitator  – if would be good to add additional details to this figure to reflect this as you have done with engagement and networks etc…

Thank you for your feedback and for addressing the points related to Figure 3 and Table 2.

We have updated Figure 3 to ensure that all constructs, including "interdisciplinary," are consistently used throughout. We have also clarified what "occasional" refers to in the context of the network and communication barrier. Specifically, "occasional" refers to specific instances where communication challenges were noted, such as during team formation, rather than being a persistent issue throughout the initiative.

Additionally, we have provided further explanations for why certain constructs, such as "culture," appear as both barriers and facilitators. This dual categorization reflects the nuanced feedback from participants, where these constructs were identified as both supportive and challenging, depending on the context and specific circumstances.

Furthermore, we addressed your point about having "Implementation Climate" and "Readiness for Implementation" as both barriers and potential facilitators in Table 2 but not in Figure 3. This is explained by explaining that the numbers in higher-level constructs (that have more detailed constructs beneath them) are aggregated, and thus, these higher-level constructs are not, by themselves, considered.

These changes should enhance the clarity and accuracy of both Figure 3 and the associated explanations, providing a more complete and coherent representation of the study's results. Thank you for your constructive feedback.

Comment 17: Line 983 – for supplementary material, include reference to all supplementary docs now

Thank you for the suggestion to reference all supplementary documents. We have now included references to all supplementary materials to ensure that readers can easily locate and access information in these additional resources. We appreciate your guidance on this matter.

Comment 18: Subdomain of CFIR mentioned in discussion but nowhere else – what are you referring to ?

Thank you for pointing that out. We have removed the reference to the subdomain of CFIR in the discussion to ensure consistency throughout the manuscript. This adjustment helps maintain clarity and prevents any confusion for the readers. We appreciate your attention to detail.

Thank you for your thorough review and valuable feedback on our manuscript. Your insights have contributed significantly to improving our work's clarity, coherence, and overall quality. We appreciate the time and effort you have dedicated to this process and are confident that the revisions we have made address your concerns. We are optimistic that these changes will meet your expectations and look forward to your final assessment.